



# Merging regional and global AOD records from 15 available satellite products

Larisa Sogacheva[1], Thomas Popp[2], Andrew M. Sayer[3,4], Oleg Dubovik[5], Michael J. Garay[6], Andreas Heckel[7], N. Christina Hsu[8], Hiren Jethva[3,4], Ralph A. Kahn[8], Pekka Kolmonen[1], Miriam Kosmale[2], Gerrit de Leeuw[1], Robert C. Levy[8], Pavel Litvinov[9], Alexei Lyapustin[8], Peter North[7], Omar Torres[10]

[1] Finnish Meteorological institute, Climate Research Program, Helsinki, Finland
[2] German Aerospace Center (DLR), German Center for Remote Sensing (DFD), Oberpfaffenhofen, Germany
[3] Goddard Earth Sciences Technology And Research (GESTAR), Universities Space Research Association, Columbia, MD, USA
[4] NASA Goddard Space Flight Center, Greenbelt, MD, USA
[5] Laboratoire d'Optique Atmosphérique, CNRS – Université Lille, France
[6] Jet Propulsion Laboratory, California Institute of Technology, Pasadena, CA, USA
[7] Dept. of Geography, Swansea University, Swansea UK
[8] Climate and Radiation Laboratory, Earth Science Division, NASA Goddard Space Flight Center, Greenbelt, MD 20771, USA
[9] Generalized Retrieval of Atmosphere and Surface Properties SAS, Lille, France
[10] Atmospheric Chemistry and Dynamics Laboratory, Earth Science Division, NASA Goddard Space Flight Center, MD 20771, USA.

*Correspondence to*: Larisa Sogacheva (larisa.sogacheva@fmi.fi)

**Abstract.** Satellite instruments provide a vantage point to study aerosol loading consistently over different regions of the world. However, the typical lifetime of a single satellite platform is on the order of 5-15 years; thus, for climate studies the usage of multiple satellite sensors should be considered. This paper assesses some options for creating merged products from an ensemble of 15 individual aerosol optical depth (AOD) data records produced from a broad range of institutions, sensors, and algorithms.

Discrepancies exist between AOD products due to differences in their information content, spatial and temporal sampling, calibration, retrieval algorithm approach, as well as cloud masking and other algorithmic assumptions. Users of satellite-based regional AOD time-series are often confronted with the challenge of choosing the appropriate dataset for the intended application. In this study AOD products from different sensors and algorithms are discussed with respect to temporal and spatial differences.

Several approaches are investigated to merge AOD records from different satellites, based on evaluation and inter-comparison results. Global and regional comparison of AOD monthly aggregates with ground-based AOD from the Aerosol Robotic Network (AERONET) indicates that different products agree qualitatively for major aerosol source regions on annual, seasonal and monthly time scales, but have regional offsets. All merged regional AOD time series show highly consistent temporal patterns illustrating the evolution of regional AOD. With few exceptions, all merging approaches lead to similar results, reassuring the usefulness and stability of the merged products.



Here, we introduce a monthly AOD merged product for the period 1995-2017, which provides a long-term perspective on AOD changes over different regions of the world. We show that the quality of the merged product is as good as that of individual products. Optimal agreement of the AOD merged product with the AERONET further demonstrates the advantage of the merging of multiple products.

**1 Introduction**

Interactions of atmospheric aerosols with clouds and radiation are the largest source of uncertainty in modelling efforts to predict climate change (IPCC, 2018).  To reduce such uncertainties, we need observations to constrain these models. However, these observations must be accurately calibrated and validated, have consistent or at least well-characterized uncertainties, and provide adequate temporal and spatial sampling over a long period of time.

With their ability to cover the globe systematically, satellites have produced major advances in our understanding of the climate system and its changes, by retrieving the spatio-temporal states of the atmosphere, land and oceans, and aspects of the underlying processes.

However, since the typical lifetime of a single satellite platform is on the order of 5-15 years, a single sensor data record may not be long enough to discern a climate signal (WMO, 2017).  Moreover, aerosol products from single satellite sensors are

often not sufficient due to limited spatial and temporal coverage and the need to avoid contamination by clouds. Thus, the application of satellite observations for climate change studies requires using products from multiple sources.

The key parameter used for various aerosol-related studies is the aerosol optical depth (AOD), which is the vertical integral of extinction by aerosol through the atmospheric column. Over the last several decades, remote sensing of AOD from space has been performed using a wide variety of sensors having different characteristics: passive and active, ultraviolet (UV) to

thermal infrared (TIR) spectral regions, single-view to multi-view, single-pixel to broad swath, sub-km to tens of km resolution, intensity-only and polarimetric, different orbits and observation time(s). Table 1 lists the data sets used in this study together with key references. Except for EPIC (orbiting on DSCOVR in L1 orbit), all other sensors are in polar-orbiting sun-synchronous low-earth orbits (~600-800 km). Only a few of these sensors were optimized for accurate retrievals of aerosols properties, and for many, AOD at one or more visible wavelengths is the only quantitatively-reliable parameter

they provide. Thus, we expect significant differences in AOD products retrieved from those sensors. Table 1 is not exhaustive; other platforms include active sensors such as the Cloud-Aerosol Lidar with Orthogonal Polarization (CALIOP) and imaging radiometers on geostationary satellites. These have very different sampling characteristics (i.e. CALIOP profiles a line, with areas either viewed twice daily and twice during the night during a month, or not at all; geostationary sensors sample a constant disk with a typical frequency of 10 minutes to 1 hour); this means their monthly mean products are

conceptually very different from polar-orbiters, so they are not considered here.

Even with one algorithm, using common basic principles applied to several instruments with similar but not identical characteristics, differences between products exist (Sayer et al., 2017, 2019; Li et al., 2016b). Further, even between the





MODIS Terra and Aqua sensors, essentially identical instruments to which the same AOD retrieval algorithm is applied, there remain differences due to calibration and time-of-day differences between the sensors (Sayer et al., 2015; Levy et al., 2018). Using different retrieval algorithms between products introduces additional discrepancies (Kokhanovsky and de Leeuw, 2009; Kinne, 2009). Likewise, different algorithms applied to the same data set, such as the three algorithms applied

to AATSR, provide similar but slightly different results (de Leeuw et al., 2015; Popp et al, 2016). Retrieval assumptions may work well in certain conditions, but not globally. Thus, regional differences in the consistency between AOD products exist (Li et al., 2014b). An important factor behind the differences could be related to the strictness of cloud masking, affecting which pixels are processed by retrieval algorithms, propagating into differing levels of clear-sky bias in daily and monthly aggregates (Sogacheva et al., 2017; Zhao et al., 2013; Li et al., 2009). Escribano et al. (2017) estimated the impact of

choosing different AOD products for a dust emission inversion scheme and concluded that the large spread in aerosol emission flux over the Sahara and Arabian Peninsula is likely associated with differences between satellite datasets. Similarly, Li et al. (2009) concluded that differences in cloud-masking alone could account for most differences among multiple satellite AOD datasets, including several for which different algorithms were applied to data from the same instrument. Due to these discrepancies, none of the satellite AOD products gives identical values of aerosol properties or is

uniformly most accurate (de Leeuw et al., 2015, 2018; Kinne et al., 2006). In other words, there is no single "best" AOD satellite dataset globally. However, different techniques applied to reveal the spatial and temporal differences between AOD monthly products, e.g. principal component analysis (Li et al., 2013; Li et al., 2014b) or maximum covariance analysis (Li et al., 2014a, b), show that there are key similarities among the AOD products tested, including MODIS, MISR and SeaWiFS. Merging multi-sensor AOD products holds the potential to produce a more spatially and temporally complete and accurate

AOD picture. With multiple observational datasets available, it is important to examine their consistency in representing the aerosol property variability in these dimensions, which is useful for constraining aerosol parametrizations in climate models (Liu et al., 2006), in the study of aerosol climate effects (Chylek et al., 2003; Bellouin et al., 2005) and for verifying global climate models (e.g., Kinne et al., 2003; 2006; Ban-Weiss et al., 2014), where satellite-retrieved AOD monthly aggregates are used.

However, to integrate a collection of several satellite aerosol products into a coherent and consistent climatology is a difficult task (Mishchenko et al., 2007; Li et al., 2009). There are only few studies, where an AOD record was merged from different satellites. Chatterjee et al. (2010) describe a geostatistical data fusion technique that can take advantage of the spatial autocorrelation of AOD distributions retrieved from MISR and MODIS, while making optimal use of all available data sets. Tang et al. (2016) performed a spatio-temporal fusion of satellite AOD products from MODIS and SeaWiFS using

a Bayesian Maximum Entropy method for East Asia and showed that, in the regions where both MODIS and SeaWiFS have valid observations, the accuracy of the merged AOD is higher than those of the MODIS and SeaWiFS AODs alone. Han et al. (2017) improved the AOD retrieval accuracy by fusion of MODIS and CALIOP data. Sogacheva et al. (2018b) combined ATSR and MODIS AOD to study the trends in AOD over China during the period 1995-2017.



Naeger et al. (2016) combined daily AOD products from polar-orbiting and geostationary satellites to generate a near-real-time (NRT) daily AOD composite product for a case study of trans-Pacific transport of Asian pollution and dust aerosols in mid-March 2014. Li et al. (2016a) constructed a monthly mean AOD ensemble by combining monthly AOD anomaly time series from five widely used satellite products (MODIS, MISR, SeaWiFS, OMI and POLDER) and applying an Ensemble

Kalman Filter technique to these multi-sensor and ground-based aerosol observations to reduce uncertainties. Penning de Vries et al., (2015) examined relationships between monthly mean aerosol properties (AOD and extinction Ångström exponent from MODIS, UV Aerosol Index from GOME-2) and trace gas column densities and showed the advantage of using multiple datasets with respect to the characterization of the aerosol type. Boys et al. (2013) combined SeaWiFS and MISR AOD data with the GEOS-Chem global model to create and study trends in a 15-year time series of surface particulate

matter levels.

When merging datasets, clearly identifying the limitations of each one should be considered. Taking advantage of the strengths of single sensors when merging AOD products, derived from different satellite instruments, could help move toward the goal of a long-term, consistent, community AOD record. On the other hand, the spread of satellite AOD records also contains added value for constraining the uncertainty of the satellite knowledge. While a lack of diversity among data

sets does not mean that they have converged on the true value, the existence of unexplained diversity does imply they have not. Note that, as with all measurements, even the AERONET spectral AOD measurements, which we adopt as the evaluation standard, have limitations. For example, AERONET includes ~450 active stations in 2019, offering far more spatial coverage than in 1993, when the network was founded, yet even now, AERONET spatial sampling is very limited for the current application, especially in regions where aerosol gradients are large, e.g., near sources (e.g., Li et al., 2016a).

To assess their consistency, the products should be compared during overlapping periods, because interannual and shorter-term variability can be significant in some parts of the world (e.g. Lee et al. 2018). In the current study, AOD monthly aggregates from different products were evaluated with ground-based measurements such as those from the Aerosol Robotic Network (AERONET, Holben et al., 1998). Based on the comparison with AERONET, we estimate how well the satellite AOD monthly aggregates reproduce the AERONET AOD climatology. To reveal the spread among AOD products globally,

we consider areas with different aerosol types, aerosol loading and surface types, which are the dominant factors affecting AOD product quality. Considering different regions globally, we also identify the strengths and weaknesses of the aggregate dataset in capturing different aerosol conditions, and the performance of the individual aerosol retrieval algorithms over different surface types. This allows users to choose the AOD product of better quality, depending on the area and research objective. A verification of open-ocean monthly data using the Maritime Aerosol Network (MAN, Smirnov et al., 2009) is

not possible in this way, because MAN data are acquired during cruises on ships of opportunity rather than as regular, repeating observations at specific locations.

Different methods for merging the AOD products (mean, median, shifted, weighted according to evaluation results) are introduced in the current paper. AOD evaluation results are used to merge the L3 monthly AOD data globally for the period



1995-2017. The AOD merged datasets are evaluated against AERONET globally and regionally, and for each AOD product separately. Annual, seasonal and monthly regional time series obtained with different merging methods are compared.

This study grew out of discussions at annual AeroSat (https://aerosat.org, last accessed 09.05.2019) meetings about how to move forward on the difficult topic of combining distinct aerosol data records. AeroSat is a grass-roots group of several

dozen algorithm developer groups and data users, meeting in person around once a year in concert with its sibling AeroCom group of aerosol modelers (https://aerocom.mpimet.mpg.de/index.php?id=2404, last accessed 09.05.2019), to highlight developments and discuss current issues in the field of satellite aerosol remote sensing.

The paper is organized as follows. The main properties of the instruments and basic principles of the AOD retrieval algorithms are summarized in Sect. 2. In Section 3, AOD datasets and regions of interest are introduced. Main principles for

the statistical evaluation of individual monthly AOD retrievals are presented in Sect. 4 (and detailed results are contained in the Annex). Alternative methods for merging are discussed in Sect. 5 and as a result, annual, seasonal and monthly AOD time series are presented and discussed in Sect. 6 and 7.  A brief summary and conclusion are given in the final section.

## 2 Instruments and algorithms/products

An overview of the instruments and AOD products included in this study is presented in Table 1. AOD products from the

same instruments retrieved with different algorithms are named in the paper with the instrument and retrieval algorithms, e.g., ATSR ADV, ATSR SU, Terra Dark Target (DT) & Deep Blue (DB) and Terra MAIAC. When both Terra and Aqua are considered, we call them together as MODIS DT&DB or MODIS MAIAC. Note that we used the merged MODIS Deep Blue and Dark Target product (denoted "DT&DB"), rather than the results of the individual DB and DT algorithms, as this merged dataset was introduced into the product for purposes like the one explored in this work.

## 3 AOD data and regions of interest

### 3.1 Monthly AOD aggregates

AOD L3 (1°x1° resolution) monthly global products were utilized in the current study. The MISR Standard L3 AOD product (0.5°x0.5° resolution) was aggregated to 1° to match the other datasets by simple averaging. Note that because of differences in instrument capabilities and swath widths, the spatial and temporal coverage of the data for calculating the monthly

average are quite different among the satellite products. Some datasets provide measures of internal diversity (e.g., standard deviation), but none currently provides estimates of the monthly aggregate uncertainty against some standard. This is an area currently being investigated by the AeroSat group due to the wide use of the L3 products.

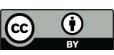



**Table 1.** Overview of the sensors, data records and AOD algorithms discussed in this paper.

| Sensor(s) | Coverage, Resolution | Algorithm Version | Algorithm Principles | References |
|---|---|---|---|---|
| Total Ozone Mapping Spectrometer (**TOMS**) (*UV spectrometer*) | 1978–1993, 1996–2003, 1° × 1° daily/monthly | Nimbus-7/TOMS: N7AERUV Ver. 0.4.3, EP/TOMS: EPAERUV Ver. 0.1.3. | Enhanced sensitivity of TOA spectral reflectance to aerosol extinction and absorption | Torres et al. (1998, 2005) |
| Advanced Very High Resolution Radiometer (**AVHRR**) (*Bispectral, single-view, broad-swath radiometer*) | 1989-1991 (NOAA7), 1995-1999 (NOAA14), 2006-2011 (NOAA18), 8.8 km, 0.5° and 1°, daily and monthly | Deep Blue/SOAR, V. 4 | Land: surface modeled using data base or NDVI. Water: bispectral simultaneous retrieval | Hsu et al. (2017) Sayer et al. (2017) |
| Along-Track Scanning Radiometer (**ATSR-2**) and Advanced ATSR (**AATSR**), both called as **ATSR** (*dual view radiometer in the visible and near-infrared; thermal infrared for cloud*) | 1995–2003, 2002–2012, global coverage in ~ 6 days,10 km daily 1° daily and monthly | ADV/ASV V2.31 | Land: spectral constant reflectance ratio Ocean: modelled reflectance | Flowerdew and Haigh (1995) Veefkind et al. (1998) Kolmonen et al. (2016) Sogacheva et al. (2017) |
| | | SU V4.3 | Iterative model inversion for continuous retrieval of AOD and FMF. Land: retrieval of BRDF parameters. Ocean: prior reflectance model. | North et al. (1999) North 2002 Bevan et al. (2012) |
| | | ORAC V4.01 (in current paper, not as separate product but as a part of the ATSR ensemble) | Optimal estimation Land: SU surface parametrization Ocean: sea surface reflectance model | Thomas et al. (2009) Sayer et al. (2010) |
| | | ATSR ensemble V. 2.7 | Uncertainty weighted mean of ATSR2/AATSR baseline algorithms ADV, ORAC and SU | Kosmale et al., in prep. |
| Sea-viewing Wide Field-of-view Sensor (**SeaWiFS**) (*Multispectral, single-view, broad-swath radiometer*) | 1997-2010, 13.5 km, 0.5° and 1° daily and monthly | Deep Blue/SOAR V.1 | Land: surface modeled using data base or NDVI. Water: multispectral simultaneous retrieval | Sayer et al. (2012a, b) Hsu et al. (2004, 2013) |
| Multiangle Imaging SpectoRadiometer (**MISR**) (*Multispectral (4-band; Vis-* | 380 km swath; global coverage ~ once /week; 4.4 km | Standard Product (SA) V23 | Land: surface contribution estimated by empirical orthogonal functions (EOFs) and assumption | Martonchik et al. (2009) Garay et al. (2017) Witek et al. (2018) |



Atmospheric Chemistry and Physics Discussions — Open Access — EGU

| Instrument | Product | Coverage | Method | References |
|---|---|---|---|---|
| NIR), multiangle (9-angle) radiometer) | | | of spectral shape invariance Water: Two-band (red, NIR) retrieval with known surface wind speed using cameras not affected by sun glint Both: Lookup table with 74 mixtures of 8 different particle distributions | Kahn et al. (2010) Garay et al. (2019) |
| Moderate Resolution Imaging Spectroradiometer (MODIS) Terra and Aqua (Multispectral, single-view, broad-swath radiometer) | DT&DB C6.1 | Terra: 2000-present Aqua: 2002-present 2300 km swath global coverage ~ two days, 10 km and 1°, daily, 8-day, and monthly | DT: Lookup table approach, surface is "known" function of wind speed (ocean), or parameterized relationship at different wavelengths (land – vegetation/dark soil) DB: Lookup table approach, climatology of surface reflectance | DT: Levy et al. (2013, 2018) Gupta et al. (2016) DB: Hsu et al. (2013, 2019) DT&DB: Levy et al. (2013) Sayer et al. (2014) |
| | MAIAC V6 | | Simultaneous retrieval of surface and aerosol from time series of observations | Lyapustin et al. (2018) |
| Ozone Monitoring Instrument (OMI) (UV spectrometer) | OMAERUV V.1.8.9.1 | 2004– 2016 1° daily, monthly | Enhanced sensitivity of TOA spectral reflectance to aerosol extinction and absorption | Jethva and Torres (2011), Torres et al. (2007, 2013, 2018) |
| Polarization and Directionality of the Earth's Reflectances (POLDER) 3 (Multispectral, multiangle polarimeter) | GRASP V.1 | Dec 2004 – Dec 2013 Global coverage in ~two days, swath 2100 ×1600 km, 5.3 ×6.2 km at nadir, and 1° daily, monthly, and seasonally | Simultaneous retrieval of surface and aerosol in frame of multi-pixel approach: statistically optimized fitting of large groups of pixels | Dubovik et al. (2011, 2014, 2019) |
| Visible Infrared Imaging Radiometer Suite (VIIRS) (Multispectral, single-view, broad-swath radiometer) | Deep Blue/SOAR, V.1 | 2012-present, 6 km and 1°, daily, and monthly | Land: surface modeled using data base or spectral relationship. Water: multispectral simultaneous retrieval | Sayer et al (2018a, b, 2019) Hsu et al (2019) |
| Earth Polychromatic Imaging Camera (EPIC) (Multispectral radiometer orbiting at Lagrange point) | MAIAC V1 | 2015-2016, 10 km | Simultaneous retrieval of surface and aerosol from time series of observations | Huang et al. (2019) |



For the inter-comparison between AOD products, three "reference" years were chosen:

- 2000, when the AOD products from TOMS, AVHRR, SeaWiFS, ATSR2, MODIS Terra and MISR are available (for the full year, except for MISR and MODIS Terra, which were available from March to December);

- 2008, when the AOD products from AATSR, MODIS Terra and Aqua, MISR, AVHRR, SeaWiFS and POLDER are

available;

- 2017, when the AOD products from MODIS Terra and Aqua, MISR, VIIRS and EPIC are available;

For products with incomplete or no coverage over ocean (TOMS, OMI, and MAIAC-types products (Terra MAIAC, Aqua MAIAC, EPIC)), the AOD over land product was considered only.

## 3.2 Regions of interest

This study focuses on different regions across the globe, as regional differences in AOD loading, types, seasonality, surface reflectance exist (Holben et al., 2001; Dubovik et al., 2002; Pinty et al., 2011), which can affect the retrieval regional quality considerably. As such, applications drawn from the products will be analysed on a regional level.

To study regional differences over the globe, we chose 15 regions that seem likely to represent different aerosol/surface conditions (Fig. 1). There are 11 land regions: Europe (Eur), Boreal (Bor), Northern, Eastern and Western Asia (AsN, AsE

and AsW, respectively), Australia (Aus), Northern and Southern Africa (AfN and AfS), Southern America (AmS), east and west of Northern America (NAE and NAW), two regions over ocean: Saharan dust outbreak over the central Atlantic (AOd) and possible biomass burning outbreak over southern Atlantic (AOb), and one region, Indonesia (Ind), that includes both land and ocean. Furthermore, we studied AOD over all land, all ocean and globally, when observed. South-eastern China (ChinaSE), which is part of the AsE region, was also considered separately as an area with considerable AOD changes

during the last 25 years (Sogacheva et al., 2018a, 2018b). Altogether, we consider the AOD in 18 regions.

For six regions, Eur, ChinaSE, AfN, AOd, Ind, AOb that differ considerably in AOD loading/type and surface characteristics, as well as for land and ocean globally, the results are presented and discussed in the main part of the manuscript. The results for the other 10 regions are shown in the Annex and briefly discussed there.

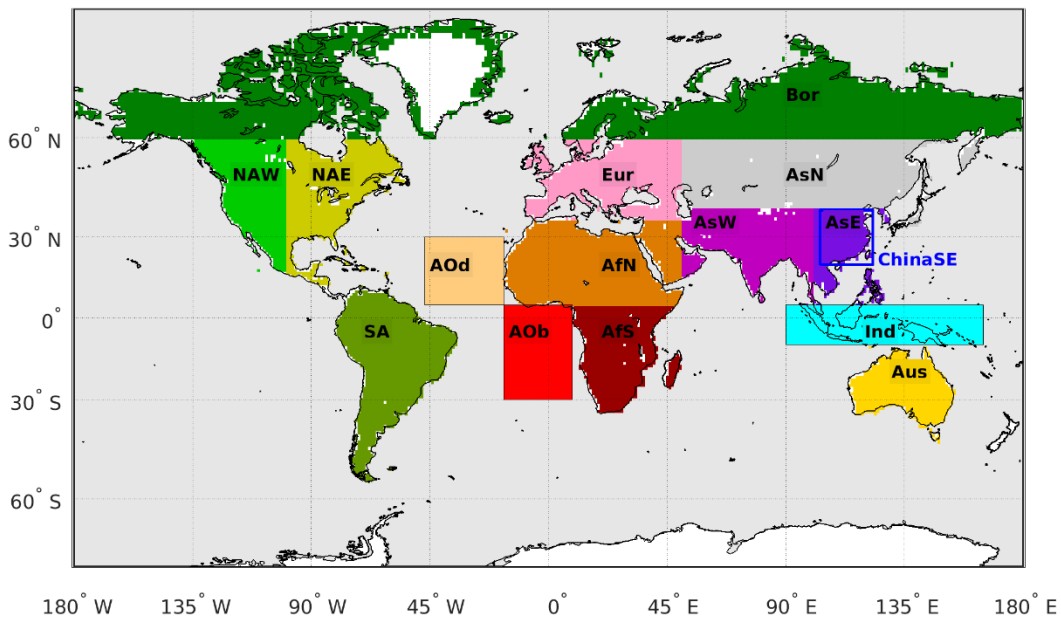

**Figure 1: 15 Land and ocean regions defined in this study: Europe (Eur), Boreal (Bor), northern Asia (AsN), eastern Asia (AsE), western Asia (AsW), Australia (Aus), northern Africa (AfN), southern Africa (AfS), South America (SA), eastern North America (NAE), western North America (NAW), Indonesia (Ind), Atlantic Ocean dust outbreak (AOd), Atlantic Ocean biomass burning**
**outbreak (AOb). In addition, Southeast China (ChinaSE), which is part of the AsE region, marked with a blue frame, is considered separately. Land, ocean and global AOD were also considered.**

## 4 AOD products inter-comparison and evaluation with AERONET

### 4.1 AOD spatial distribution

To reveal spatial differences among the annual AOD products, annual offsets were estimated from the median AODs of all
10 available products (Fig. 2, upper panel). Year 2008 was chosen for this exercise, as AOD data are available from most instruments (Sect. 3.1). For TOMS, which is not available in 2008, the annual AOD difference from the median AOD was calculated for year 2000; for VIIRS and EPIC 2017 was chosen for the inter-comparison. Median AOD is slightly higher in 2008 over both land (0.184) and ocean (0.135), and thus globally (0.150), compared with 2000 and 2017 over land (0.180/0.174), ocean (0.128/0.132) and globally (0.143/0.145), respectively. However, all these differences remain below
15 0.010. Although AOD is higher over ChinaSE in 2008 compared to other years, similar AOD spatial patterns are observed for the 3 years chosen for the inter-comparison. Thus, using those 3 years (2000, 2008, 2017) to compare global, annual-averaged AOD deviation from the median AOD, is suitable for revealing differences among the products.

Over land, TOMS AOD is about twice as high (by 0.162) as the median land AOD for all available products (0.180); OMI is higher by 0.06. One explanation for such high AOD differences for TOMS and OMI is possible cloud contamination, related





to the lower pixel resolution (~ 50 x 50 km-square) compared to the other instruments, (Sect 2). Global AOD is also higher for POLDER by 0.049, over both land (by 0.096) and ocean (by 0.029), except for the dust area AfN, were AOD is lower than the median.

Products from different instruments, retrieved with similar algorithms (AVHRR, SeaWiFS, VIIRS), often show similar
results. AVHRR and SeaWiFS tend to underestimate AOD compared to the median, whereas VIIRS global AOD provides only a small overestimation (by 0.006) of the median value. MODIS DT&DB show higher AOD compared to the median; the deviation of Terra DT&DB from the median (by 0.027 and 0.031 for land and ocean, respectively) is about twice as high as that for Aqua DT&DB (by 0.013 for both land and ocean). This offset between the two MODIS sensors has been reported previously; although time-of-day differences might contribute, it is thought to be dominated by calibration offsets, and has
improved compared to previous data versions (Levy et al., 2018; Sayer et al., 2019).

Interestingly, different algorithms applied to the same instrument, often show opposite results, providing an algorithmic spread from the median value. Unlike MODIS DT&DB, MODIS MAIAC AOD are both lower than the median over land (by 0.038 and 0.044, respectively). ATSR ADV is slightly higher over ocean (by 0.026), whereas ATSR SU is lower than the ocean median by a similar amount (0.024). Over land, ATSR ADV is lower, especially over bright surfaces (Sect. 2.6.1), if
retrieved, whereas ATSR SU is considerably higher over bright surfaces, compared to the median value. As a result, ATSR ADV AOD underestimates (by 0.011) and ATSR SU overestimates (by 0.026) the median over land.

Although the global land and ocean AOD values are similar (within ± 0.03 of the median AOD for most products, which is inside the GCOS requirement of the greater of 0.03 or 10%; GCOS, 2016), contrasting AOD behavior appears in several regions.  In ChinaSE, OMI, AVHRR, SeaWiFS, VIIRS and MISR AOD are considerably lower (up to -0.25) than the
median AOD, whereas the MODIS-family (Terra DT&DB, Aqua DT&DB, Terra MAIAC, Aqua MAIAC) retrieve up to 0.2 higher AOD than the median. Over bright surfaces (e.g., AfN, AsW), the OMI, AVHRR, VIIRS, ATSR SU, ATSR ensemble retrieves up to 0.2-0.25 higher AOD than the median, whereas the MAIAC-family (Terra MAIAC, Aqua MAIAC, EPIC) and POLDER AOD are lower by up to 0.1-0.15 AOD than the median. Over open ocean, the VIIRS and ATSR ensemble show AOD closest to median among the products. As there is no clear deviation from the median for almost all products
over open ocean in the continental outflow areas (e.g. AOd and AOb), this confirms that the datasets contain similar AOD in these regions.  The phenomenon of high Southern Ocean AOD is found in several satellite data sets, including MODIS Terra and Aqua DT, MISR and POLDER and may be related to unresolved clouds (e.g., Toth et al., 2013; Witek et al., 2018).





**Figure 2. Upper line: annual AOD median for 2000 (\*), 2008 and 2017 (X), calculated from the available products. Lines 2-6: AOD deviation of the different products from the annual median AOD for years 2000 (TOMS), 2017 (VIIRS and EPIC) or 2008 (all other products). AOD anomalies with respect to the AOD median are shown on the deviation plots. Global land and ocean AOD mean differences are shown for each product, when available. For summer, see Fig. A1.**



Seasonal deviations from the median AOD are similar to the annual patterns throughout the year. However, the spread among the products is slightly more pronounced in summer (Fig. A1 for JJA, summer for the Northern Hemisphere), when the absolute AOD often reaches its maximum in certain regions (e.g., in China, Sogacheva et al., 2018).

Thus, the AOD deviations from the median show regional differences, even for products retrieved from the same instruments with similar algorithms. As both negative and positive deviations are observed in regions with high AOD loading, the surface type is also likely to influence the AOD retrieval. High AOD loading might, in turn, be wrongly screened as cloud and thus bias monthly AOD lower. As with many factors that contribute significantly to AOD retrieval results, surface treatment and cloud screening should be tested with L2 (higher-resolution, swath-based resolution) daily data; but this is

beyond the scope of the current study, which considers only L3 monthly AOD products.

To further reveal differences among the AOD products retrieved with different algorithms applied to different satellites, the diversity of the satellite annual mean AOD ($AODdiv$) was calculated, as in Chin et al., (2014):

$$AODdiv = 0.5 \frac{AODmax - AODmin}{AODmedian} * 100\%$$

The seasonal AOD diversity was calculated for the test years 2000, 2008 and 2017 from all available products (Fig. 3). As

expected, satellite data agree best over ocean, and diversity decreases from 2000 (40-60%) towards 2017 (20-30%). Higher over-water diversity is often seen in high latitude oceans, for which retrievals are more challenging due to cloud cover and the high solar zenith angle, which leads to both more limited sampling and more retrieval artefacts (Toth et al., 2013). Note that the results slightly differ from Chin et al., (2014), since in our analysis, besides TOMS, AVHRR and NASA's Earth Observing System (EOS) satellites (SeaWiFS, MISR and MODIS), the European Space Agencies' AATSR radiometer on

board the Environmental Satellite (ENVISAT) is also included for years 2000 and 2008. Also, in some cases, newer versions of the satellite products than those available in Chin et al. (2014) were used here.

The diversity is considerably higher over land, reaching more than 90% over certain areas in year 2000 (NAW, SA, Siberia, central Asia, AfS and Aus). AOD diversity decreases considerably in 2017 (to 30-40% on average), when AOD from only NASA Earth Observation System (EOS) satellites (MISR, MODIS, VIIRS) is currently available, and the size of areas with

high diversity is lower. Australia stands out as the region where AOD averages disagree most, showing >90% AOD diversity irrespective of the year or number of satellites/products available. The diversity is somewhat higher in summer, which is also clear from the comparison between annual (Fig. 2) and summer (Fig. A1) AOD deviation from the median value.

As discussed early in this section, AOD for TOMS, OMI and POLDER differ most from the median AOD. Those products contribute most (up to the 40-50%) to the AOD spread (figures not included). However, we use all available products in the

current exercise to obtain the longest possible data record to be used later for trend assessment.

**Figure 3. Seasonal mean AOD diversity for years 2000, 2008 and 2017 for all available products**

## 4.2 Evaluation of monthly AOD

Verification of the AOD monthly aggregates is of great importance, since model evaluation is often performed on the
5   monthly temporal resolution (Nabat et al., 2013; Michou et al., 2015; Li et al., 2016b). To evaluate the quality of any AOD
product, the verification of the product against more accurate reference measurements is obligatory. Ground-based
measurements such as from the AERONET (cloud screened and quality assured Version 3 Level 2.0, Giles et al., 2019)
provide highly accurate measures of AOD that are widely used as ground truth for the validation of L2 satellite AOD data.





Although extensive L2 AOD validation has been performed for different aerosol products, only a few attempts have been made to evaluate AOD monthly aggregates retrieved from satellites (e.g., Li et al., 2014b, Wei et al., 2018).

Verification of monthly aggregate satellite AOD is more challenging than L2 validation. The verification exercise is not a true validation, unlike instantaneous matchups with satellite retrievals which are commonly compared with AERONET.

Deviations between monthly aggregates and AERONET monthly aggregates are expected, based, e.g., on differences in satellite spatial and temporal sampling (Sec. 4.1). This issue is more significant for satellites with lower coverage and can result in missing extreme AOD events. Differences in cloud screening affect mainly high AOD events that can be erroneously removed in some products. Both coverage and cloud-screening issues can decrease monthly aggregated AOD.

By performing the verification of the AOD monthly aggregates, we aim to reveal how well different monthly satellite

products meet monthly AOD values observed with AERONET. For that purpose, we compare AOD monthly aggregates from all available data from both AERONET and each satellite product.

Results from this comparison have limitations, since AERONET provides the data over sertain locations within a grid cell, while satellites cover a larger fraction of the area of a grid cell (depending on sampling and cloud cover). So, for example, AERONET is likely to miss extreme high values (localized plumes), which can produce low biases, unless a station happens

to be directly under an aerosol plume – in which case the value there might be skewed high. A related issue for comparing satellite-retrieved monthly AOD aggregates with ground-based AERONET AOD is the spatial representativeness of the AERONET stations for chosen regions more generally (Shi et al., 2011; Li et al., 2014a, 2016; Virtanen et al., 2018). Note, that both AERONET and satellite monthly AOD aggregates are not "true" monthly AOD values, as AOD is not measured/retrieved under cloudy conditions or across the full diurnal cycle by either technique; here "AOD monthly

aggregate" means the daytime, cloud-free AOD monthly aggregate.

In the evaluation exercise, AERONET monthly mean AOD and Ångström exponent (AE, which describes how AOD depends on wavelength and is sometimes used, together with AOD, to constrain aerosol type) were calculated from AERONET daily means. Verification was performed for all available AERONET data, and separately for different aerosol types: background aerosol (AOD<0.2), fine-dominated (AOD>0.2, AE>1) and coarse-dominated (AOD>0.2, AE<1) aerosol.

These thresholds are somewhat subjective, but this simple categorization reflects typical differences between fine-mode dominated and coarse-mode dominated aerosol types (e.g. Eck et al., 1999). This classification has also been used by, e.g., Sogacheva et al. (2018a, b).

As the deviation of each satellite product from the median is regionally dependent (Fig. 2), we performed the AOD comparison between monthly aggregated products and the AERONET monthly product for each selected region separately.

Even though we tried to choose regions with homogeneous aerosol conditions, AOD conditions (and thus algorithm performance) might vary somewhat within the region, and AERONET stations might have different record lengths. To keep similar weighting for each station in a region, we first calculated statistics for each AERONET station separately, and then calculated the regional median validation statistics from all available stations.



To reveal how retrieval quality depends on AOD loading, offsets between AERONET AOD and satellite product AOD were estimated for binned AOD, and the number of observations in each AOD bin is reported. Correlation coefficient (R), offset and root-mean square error (rms), as well as the fraction of falls to the spread envelope (SE) of $\pm$ (0.05 + 0.2*AOD) and fraction of points which fulfil the GCOS requirements (GE) of 0.03 or 10% of AOD.

Monthly AOD verification results were used in this manuscript to estimate weights for each satellite dataset in some of the merging approaches. Knowledge about how well a satellite AOD record describes monthly AOD globally is also important for AOD trend estimation, as well as estimation of the aerosol impact on the global radiation balance and, thus, on climate change.

### 4.2.1 Binned offset, global evaluation

A general tendency toward positive satellite-retrieved AOD offsets (Fig. 4) is observed for most products under background conditions. Since, on average, 70-80% of monthly AOD fall into class "background" (AOD<0.2), AOD mean biases are expected to have similar behavior. TOMS, OMI and POLDER have the highest positive offsets globally, which is in line with the results from the dataset spatial inter-comparison (Sect. 4.1). Notable bias (overestimation) overall land for situations with low AOD and certain underestimations of coarse mode aerosol (desert dust), revealed during the validation of the L2

AOD of the first version of the POLDER global product is being addressed in a new retrieval version. Offsets close to zero for background AOD are observed for the MODIS MAIAC products.

For most products, except MODIS DT&DB, AOD offsets become negative for AOD>0.2 (fine- and coarse-dominated aerosol types) with increasing amplitude (up to 0.2-0.5) towards highest AOD values. MODIS DT&DB Terra and Aqua show the lowest offsets for 0.2<AOD<1. Note that the MODIS DT&DB retrieval is not entirely independent of AERONET,

as the over-land algorithm uses a region-specific aerosol-type climatology derived from AERONET measurements (Levy et al., 2013). Offsets for VIIRS are close to 0 for AOD<0.5 and reach ca. 30% of AOD at AOD≈1. For the current MISR standard product, AOD is systematically underestimated for AOD >~0.5; this is largely due to treatment of the surface boundary condition at high AOD (Kahn et al., 2010), and is addressed in the research aerosol retrieval algorithm (e.g., Limbacher and Kahn, 2019). Except for TOMS and Terra MAIAC, offsets are smaller for coarse-dominated AOD.

In summary, similar offsets (positive for AOD<0.2 and negative for AOD>0.2) may plausibly indicate systematic overestimation (by 0-0.05 for different products) of AOD<0.2 and underestimation (by more than 0.25) of high AOD (>0.2) for satellite AOD monthly aggregates. AOD products differ in their ability to meet higher AERONET monthly AOD >0.3. AOD products retrieved from satellites with better coverage show better agreement between AOD and AERONET monthly aggregates. Thus, sampling differences (swath and pixel selection) are critical, as expected. However, MODIS DT&DB

show slightly better performance than MODIS MAIAC for AOD>0.3, which might result from differences in the retrieval approach and/or cloud screening.



### 4.2.2 AOD evaluation over selected regions

Because of differences in instrument specifications and retrieval approaches, the performance of retrieval algorithms depends largely on aerosol type, aerosol loading and surface properties at certain locations (e.g., Sayer et al., 2014). In this section we evaluate AOD products by comparing statistics between each product and AERONET in four selected regions: Eur, China

SE, AfN, AOd. Results for the other 14 regions are shown in the Appendix. For each region, statistics (R, % of points in SE and GE, offset and RMSE) for all 15 products are combined into one subplot (Fig. 5 and Fig. A2). The merged AOD product M is introduced in Sect. 6.2; evaluation of that product is discussed in sect. 6.2.3.

Algorithm performance over Europe is similar for most products, with R of 0.55-0.65, 45-55% of the pixels in the GE and 70-80% of the pixels in the SE, offset of 0.05-0.1 and RMSE of ~0.1. For TOMS, OMI and POLDER the performance is

slightly worse than for other products in Europe. In ChinaSE (a region of particular interest having high AOD loading, mainly due to high levels of anthropogenic aerosols), the offset (0.1-0.2) and RMSE (0.2-0.3) are considerably higher than in Europe, and fewer pixels fit within the SE (55-70%) and GE (15-30%). This is likely due to a combination of high AOD loading and accompanied high uncertainty in the product, high variability in aerosol composition and surface properties. In Indonesia and for the biomass burning outflow over the Atlantic, the EOS products show better agreement with AERONET

than the ATSR-family products.

Products with different surface treatment (ATSR SU, MODIS-family, MISR) show similarly higher R over AfN, an area of high surface reflectance. However, a high R does not imply that the performance is better. Other statistics (number of pixels within GE, offset and RMSE) in AfN are worse compared with those in Europe.

Overall, no single product has the best statistics for all metrics and regions. Retrievals tend to perform well in areas with

darker (more vegetated) surfaces and where aerosol type is less variable over time. In these cases, biases are small and retrieval uncertainties are often better than defined SE, tracking temporal AOD variability well but with a tendency to underestimate high-AOD events. In more complex tropical environments, data should be used with greater caution, as there is a larger tendency to underestimate AOD. However, correlation often remains high, suggesting good ability to identify monthly AOD, despite this underestimation.





**Figure 4. Global AERONET comparison of the AOD monthly aggregates (circles – median bias, error bar – bias standard deviation) for all AOD types (purple), fine-dominated AOD (blue) and coarse-dominated AOD (green) and fraction of points in each bin (bar, orange)**



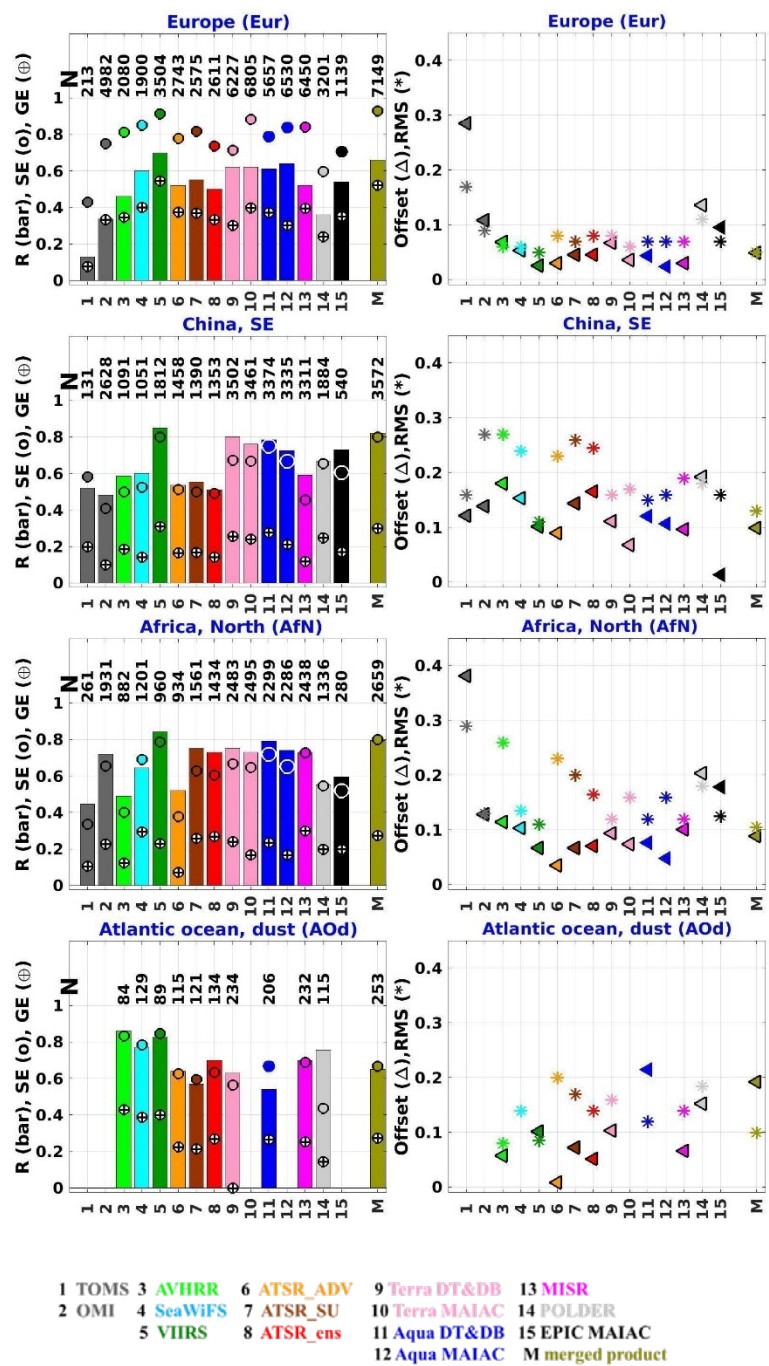

**Figure 5. AERONET comparison statistics (Left: correlation coefficient R, bar; fraction of pixels in the Spread Envelope, SE, ○; fraction of pixels satisfying the GCOS requirements, GE, ⊕; Right: Offset, △; root mean square error RMS, *) for AOD monthly**





**aggregates for each product (1:15, legend for products below the plot) and the L3 merged product (M, for details see Sect.6.2) with corresponding colors (legend) for the selected regions ( as in Fig. 1). N is a number of matches with AERONET. Note, for products which do not provide global coverage (e.g., no retrieval over oceans), the results are missing. For other regions, see Fig. A2.**

## 4.3 AOD time series

In order to move towards consistency in AOD records derived from multiple satellites using different sensors and retrieval techniques, this section examines annual regional AOD time series obtained from the different products.

Besides the positive offset for TOMS, OMI and POLDER (see Sect. 4.2), consistent temporal patterns are observed, and similar interannual AOD variability is tracked by all datasets (Fig. 6 and Fig. A2). AOD peaks in Europe in 2002, in China in 2006/2007, 2011 and 2014, (possibly related to changes in anthropogenic emissions, Sogacheva et al., 2018a, 2018b). Relative AOD peaks over the Atlantic dust area in 1998, 2012 and 2015 (Peyridieu et al., 2013), and obvious AOD peaks in Indonesia related to the intensive forest fires in 1997, 2002, 2006 and 2015 (Chang et al., 2015, Shi et al., 2019) are clearly seen.

However, significant regional offsets between products exist, which are largest in regions with high aerosol loading. Over ChinaSE, MODIS-family products show higher monthly AOD compared to all others. Over AfN, the ATSR_SU and ATSR_ensemble reach higher monthly aggregated AOD than the MODIS-family products, whereas comparisons with AERONET are similar for ATSR and MODIS (with slightly higher RMSE for ATSR by 0.05); differences are likely tied to the small number of stations in this region. A high offset between MODIS and ATSR is revealed over Australia (Fig. A3).

## 4.4 AOD annual cycles

Year 2008 was chosen for annual AOD cycle inter-comparison, when data from all products except TOMS, VIIRS and EPIC are available.

Annual AOD variations among the products (Fig. 7 and Fig. A4) agree better in regions with relatively low AOD (e.g., in Europe). In Europe, the relative increase towards summer and further increase in August-September is captured by all products. In ChinaSE, where AOD loading and annual variation are higher, the AOD increase from winter to summer and another AOD peak in September are observed. Outbreaks of the biomass burning aerosols over Atlantic produce clear peaks in February and September, as shown in all available products.

Thus, besides the similar interannual variation among the satellite products, the AOD annual cycles among the products is also similar.





**Figure 6. Annual AOD time series for the selected regions. For 2018, MODIS DT&DB and VIIRS AOD products are shown here but not used in the merging exercise. For other regions, see Fig. A3.**





**Figure 7.** Monthly (J for January, F for February, etc., left plots) and seasonal (DJF, MAM, JJA, SON, middle plots) AOD time series for the selected regions for 2008, and yearly aggregate AOD for 2008 (Y, right) for selected regions. For other regions, see Fig. A4.



## 5 AOD merging methods

We tested and compared three different approaches for merging regional and global AOD time records:
- Mean and median for the 15 individual records (approach 1)

- Mean and median of 15 offset-corrected records (approach 2)

- AOD weighted with results of the AERONET verification (approach 3)

AOD annual time series from all available products were merged for the period 1995-2017.

To achieve best estimates of the regional AOD by merging multi-sensor monthly AOD data, the uncertainties should be considered explicitly. However, this cannot (yet) be done, as currently only some of the L2 products contain pixel-level propagated or estimated uncertainties, and none of the monthly products contain anything beyond simple averages of those

L2 uncertainties.

### 5.1 Approach 1: AOD mean, median for uncorrected AOD

The mean (average) value, while a common statistic used in climate studies, is not generally equal to the most commonly occurring value (the mode) and may not be representative of the central tendency (the median) of strongly asymmetrical distributions (O'Neill et al., 2000). By itself, the mean does not provide any information about how the observations are

scattered, whether they are tightly grouped or broadly spread out. Thus, to describe the asymmetry, in addition to the mean, we also report the median and standard deviations.

### 5.2 Approach 2: AOD mean, median for shifted AOD

As shown in Sect. 4, the AOD time series of different products are highly consistent, showing similar temporal patterns. However, offsets between the products exist, which vary globally and seasonally (Fig. 6, Fig. 7). To estimate the average

offsets between the products, we chose as a reference the ATSR_ensemble product, which is closest to the median value of all products AOD (with some exceptions discussed below). As in approach 1, AOD mean, median and standard deviation were calculated from all available datasets, shifted to the ATSR_ensemble products.

### 5.3 Approach 3: Weighted AOD

As shown in Sect. 4.2, the products differ in the degree to which each represents the AERONET values on the monthly scale.

A weighted AOD mean is calculated as our third approach, by assigning weights based on their agreement with AERONET. The better-compared products are given more weight in the calculation of a combined product.

An AOD-weighted mean was calculated with a ranking approach based on the statistics from the AERONET comparison for AOD: R, bias, RMSE, GE (Fig. 5) and median bias of the binned AOD in the range [0.45, 1] (Fig. 4). The last criterion was added to specifically consider algorithm performance for higher AOD.



Two ranking methods were tested. For the first ranking method (RM1) based on best statistics, the 15 products were ranked from 1 (worst) to 15 (best) for each statistic (R, GE, RMSE, bias, binned bias) separately. The 5 separate ranks were then summed, so the maximum possible rank is 15*5=75. Possible errors in RM1 occur when several products have similar statistics, so small variations in statistics can produce large changes in ranking. Note, that no product received a perfect (75)

rating.

In the second ranking method based on binned statistics (RM2), a rank from 1 to 10 was assigned to each metric separately according to the bin number. For each statistic, the following windows: [0.5, 1] for R, [0, 0.5] for GE, [0, 0.2] for bias, [0, 0.15] for RMSE, and [-0.5, 0] for the binned bias were divided into 10 bins. In that exercise, several algorithms could be ranked similarly for certain statistics, if their statistics fell within the same bin. For example, if R for three products is

between 0.8 and 0.85, all three were ranked 8 for that statistic. If for all 15 the R was between 0.6 and 0.65, they all would receive rank 4 with RM2, whereas with the RM1 approach, they were ranked from 1 to 15, which caused an artificial bias in the ranking. RM2 is more logical, as possible errors in statistics are considered when operating with bins. For example, R might be slightly biased by a few outliers. If R is 0.82 or 0.81, the same rank of 8 is given in RM2. In RM1, the lower rank is given for the product with R= 0.815 than for the product with R= 0.82. Ranking results for RM1 and RM2 are slightly

different, but the resulting merged products are similar (not shown here). Thus, RM2 results (discussed below in Sect. 6.3) were used to determine the weight of each AOD product.

The sum of the five ranks for each product was calculated and transformed to a weight of each product (as a fraction of total sum for the product from the total sum of ranking for all products) to calculate the AOD weighted mean.

As shown in Sect. 4.2.1, the performance of the retrieval algorithms often depends on the aerosol conditions (aerosol type

and loading, Fig. 4) and surface properties. Accordingly, weights for the different AOD products were calculated separately for each region for different aerosol types considering the corresponding regional statistics from the AERONET comparison.

## 6 AOD merging results

### 6.1 Approach 1 and 2 : AOD mean, median for uncorrected and shifted AOD time series

### 6.1.1 Mean offset between AOD products and ATSR_ensemble AOD

Means and standard deviations of the offsets for all datasets from the ATSR_ensemble AOD are shown in Fig. 8 and Fig. A5 for selected regions. For VIIRS and EPIC, which do not overlap (in time) with ATSR, the offset to the ATRS ensemble was estimated by adding the offset to Terra DT&DB plus the offset of Terra DT&DB to the ATSR ensemble.

In Europe, the offset for most products falls within ±0.04 AOD. OMI, VIIRS, Terra, Aqua and EPIC are offset positive, whereas SeaWiFS, AVHRR, ATSR_ADV, ATSR_SU and MISR show negative offsets from the ATSR ensemble. TOMS

and POLDER show larger positive offsets (0.12 and 0.08, respectively) whereas MODIS MAIAC show high (ca.-0.06) negative offsets.

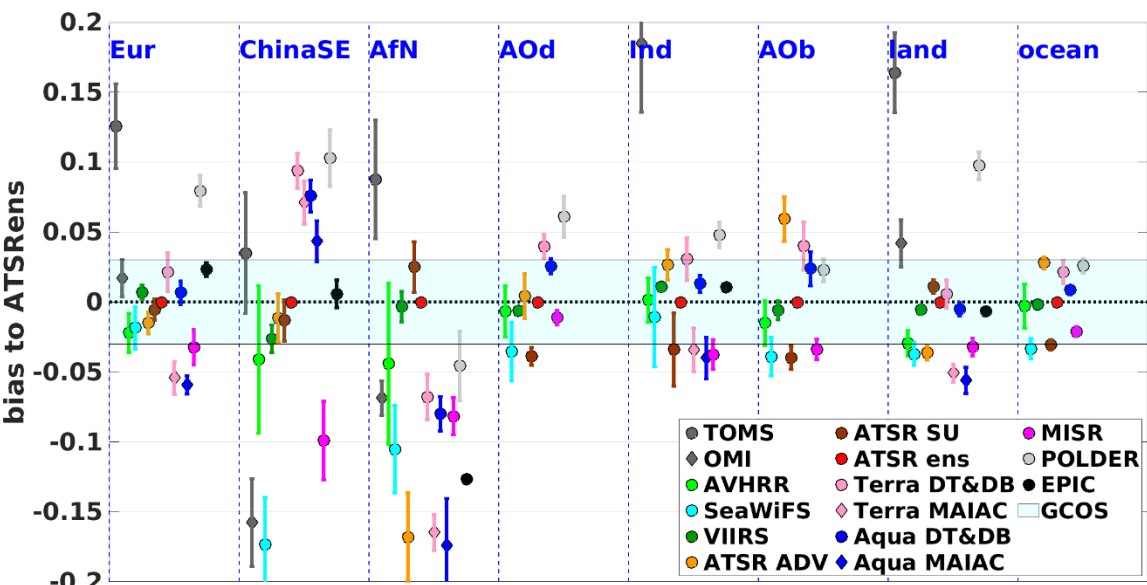

**Figure 8. Regional annual average AOD offset between each dataset and the ATSR_ensemble dataset. GCOS requirement for ±0.03 AOD is shown as a background color. For other regions, see Fig. A5.**

For most products and regions, Fig. 8 shows similar tendencies in the signs of offsets. Offset magnitudes and their variations depend on AOD loading: offsets are typically higher for high AOD. However, there are exceptions. In ChinaSE, MODIS MAIAC shows positive offsets. In AfN, most products, except TOMS and ATSR_SU, have negative offsets.

Over land, VIIRS, MODIS DT&DB have the lowest offsets (±0.01) relative to the ATSR ensemble product, whereas over

10    ocean AVHRR, VIIRS and Terra DT&DB offsets are close to zero.

With the second merging approach, each AOD product was shifted based on its regional offset with respect to the ATSR ensemble (Sect. 5.2). Mean, median and standard deviation AOD time series were then derived from these 15 shifted data records.

### 6.1.2 Mean and median AOD time series

15    Regional mean and median AOD time series calculated from all available uncorrected data records (approach 1) are shown in Fig. 9 and Fig. A6 for the period 1995-2017. Mean and median AOD time series from products shifted to the





ATSR_ensemble (approach 2) are also plotted. Products and shifted products from TOMS (1979-1993) and AVHRR (1989-1991) are shown, when available.

The two merging approaches tested here agree well except in NAf, where the ATSR_ensemble (used as reference) is biased high, as shown from comparison with AERONET, which leads to a positive bias in the shifted median compared to the

simple median of uncorrected products. The agreement of the two approaches is encouraging, as we can conclude that for the big-picture analysis of overall trends, details of the methodology do not matter so much.

There are some smaller differences between the two approaches. Time series from uncorrected records are biased high compared to the records from the shifted products, due to the high bias of TOMS, OMI and POLDER, which also leads to a larger standard deviation. By adding both MODIS products beginning in 2002, the uncorrected AOD standard deviation

becomes lower over land. For the period before 1995, where only one or two products (TOMS and AVHRR) exist, no meaningful merging can be applied; only in some regions the shifted records for this period come close to the average level of the second part of the merged records.

Using the temporal records to monitor regional AOD trends seems plausible. As demonstrated in previous studies (e.g. Zhang and Reid, 2010; Hsu et al., 2012; Chin et al., 2014), the merged records show a small but gradual decrease of AOD

over Europe (with one small peak in 2002). Spatial consistency is indicated by high correlation (similar positions of peaks) in NAfr and its Atlantic dust outflow region. Interannual variation as well as the standard deviations are highest for regions with the largest AOD, e.g., over ChinaSE (anthropogenic emissions) and Indonesia (biomass burning). The time series of ChinaSE follows the known patterns caused by step-wise regional emission reductions in the last 25 years (Sogacheva et al., 2018b).





**Figure 9. Merged time series from uncorrected AODs (approach 1, blue) and from shifted AODs (approach 2, red) for the 8 selected regions (for both: mean, dotted line, median, solid line, and 1σ shadowed area). For other regions see Fig. A6**





## 6.2 Approach 3: weighted AOD

### 6.2.1 Ranking results

Results for RM2 ranking are shown in Fig. 10 and Fig. A6 for three aerosol types (background, fine- and coarse-dominated) and all aerosol types. As with the results discussed in Sect. 4, none of the algorithms consistently outperforms the others in all regions. There is no clear leader over Europe, a region with low AOD, indicating good performance of all algorithms under background conditions. MODIS is ranked lower over the Atlantic dust region. Over land globally, the ranks are similar for EOS and ATSR, with somewhat higher number for VIIRS.

Over ocean globally, the ranks are similar for all existing products (again a region with low AOD). One reason that the VIIRS and MODIS ranks are often higher is likely their better coverage, which enables them to better represent AERONET monthly means over land. The lowest ranks are obtained consistently for TOMS, OMI and POLDER, due to their high biases.

Ranks for the different aerosol classes (all, background, fine-dominated and coarse-dominated) are different, which raises another aspect of using multiple products. Over land, MODIS MAIAC often has a higher rank for background AOD, whereas MODIS DT&DB are better for other aerosol types.



**1 TOMS  3 AVHRR    6 ATSR_ADV   9 Terra DT&DB    13 MISR**
**2 OMI   4 SeaWiFS   7 ATSR_SU   10 Terra MAIAC   14 POLDER**
**5 VIIRS     8 ATSR_ens   11 Aqua  DT&DB   15 EPIC MAIAC**
**12 Aqua MAIAC**

**Figure 10. Weight of each product for the selected regions for different aerosol types. For other regions, see Fig. A7**

### 6.2.2 AOD monthly aggregates merged based on ranking results

Using the regional weights (Fig. 10 and Fig. A7) for three different aerosol types and all types, the Approach 3 (weighted AOD) was applied to the L3 monthly aggregates of all products available since 1995. As the ranking was slightly different



for different aerosol types, merging of the L3 AOD was done for three aerosol types separately, and for all aerosol particles together when the aerosol type was not specified.

To estimate the quality of the AOD merged L3 monthly product, we performed an exercise to evaluate the merged AOD (approach 3, RM2, all aerosol types) against AERONET AOD, similar to the one used for merging all products (Sect. 4.2).

The AOD binned bias of the merged product (Fig. 11, left) shows a similarly small deviation from AERONET (±0.03) for AOD<0.5 (positive for AOD <0.3 and negative for 0.3<AOD<0.5), where the fraction of values falling within the bins is about 0.95. For AOD>0.5, where the number of cases is very low, the underestimation increases as AOD increases. For AOD<1.25, the deviation is similar for all aerosol types.

Correlation coefficient, number of the pixels in the GE, offset and RMSE for the AOD merged product are shown in Fig. 5

for selected regions. For the AOD merged product, which has the best temporal coverage, the number of points used for validation (N) is higher than for any individual product. Comparison with these metrics for the individual products in Fig. 5 shows that the correlation coefficients and the number of the pixels in the SE are as high as for the 1-2 best-ranked products in the corresponding regions. The offset is close to the averaged offset, and the RMSE tends to be lowest. Thus, the quality of the merged product is as good as that of the most highly ranked individual AOD products in each region.

The L3 merged AOD difference from the median in 2008 (Fig. 11, upper) is close to 0 over ocean (except for near the poles regions) and within GCOS requirements over land and globally (0.011 and 0.008, respectively). The statistics used to evaluate the L3 merged product against AERONET (Fig 11, lower, as in Figs. 5 and A2) combined for all 15 regions, as well as for land, ocean and globally, show that for most regions, R is between 0.75 and 0.85, 80%-90% fall within the SE, 20%-60% fall within the GE, and the RMSE and offset are between 0.05 and 0.1, though somewhat higher for the regions with

potentially high AOD loading (Indonesia and AOd, up to 0.15-0.2 for AsW and AsE).





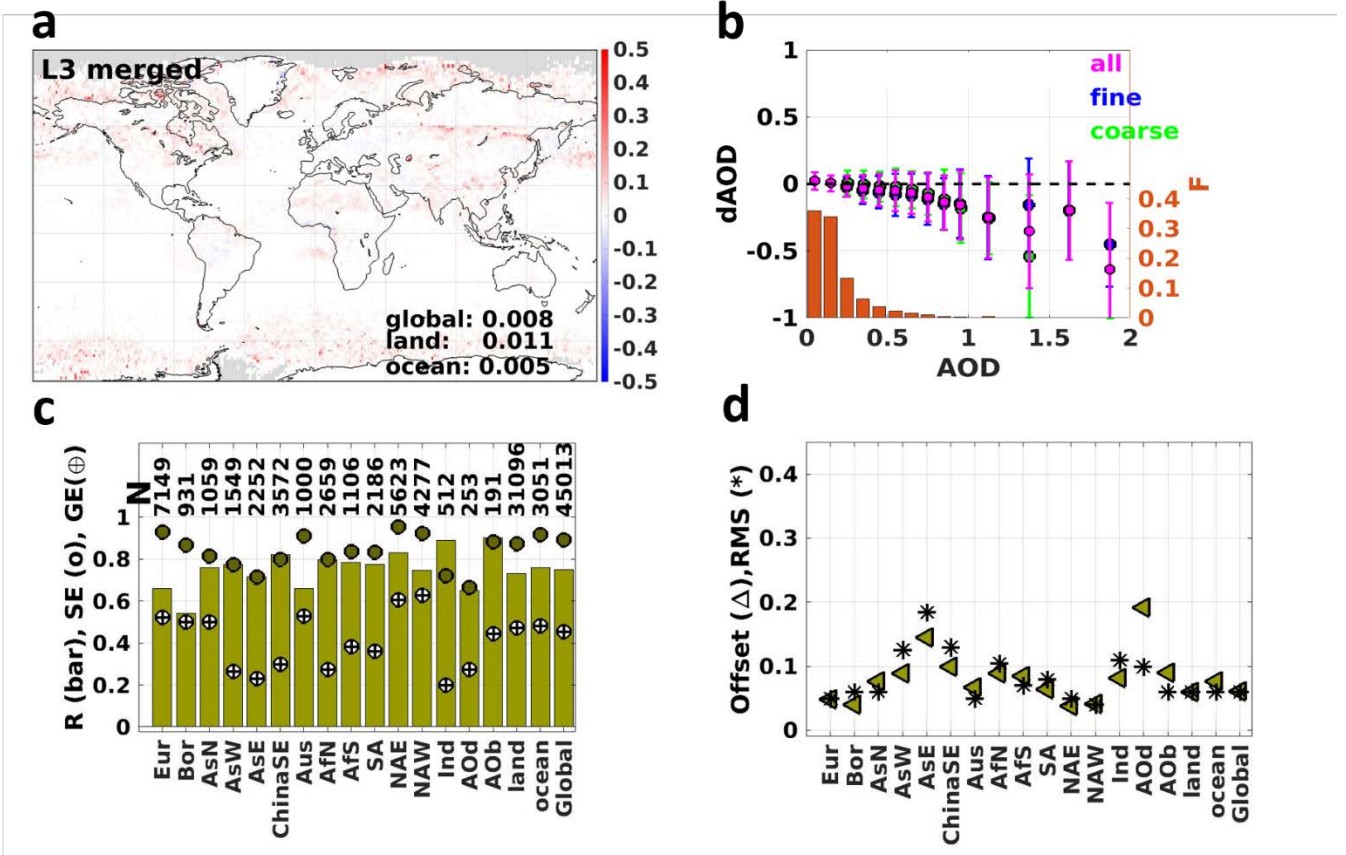

**Figure 11. L3 merged (approach 3, all aerosol types) AOD product deviation from the annual median AOD for year 2008 (as Fig. 2 for other products), a. b: L3 monthly merged AOD product evaluation with AERONET : binned AOD bias for all (purple), background (AOD<0.2, purple), fine-dominated (blue) and coarse-dominated (green), b; regional statistics (c: correlation coefficient R, bar; fraction of pixels in the Error Envelope, EE, circle; d: Offset, Δ; root mean square, star).**

### 6.2.3 AOD time series merged based on ranking results

The AOD time series produced by merging all available products, assuming the ranking for all types of aerosol particles, are shown in Fig. 12 and Fig. A7 together with the time series calculated from the L3 merged aerosol product (Sect. 6.2.2). Overall low (1-3% of AOD) 1-σ AOD standard deviation shows agreement with the time series merged for all four classes, which means that aerosol type is not critical for merging. The deviation between time series for different aerosol types increases slightly in the regions with the potentially high AOD loading. For some regions (e.g., Europe, AfN, Asia), the deviation is higher for the period when MODIS-family products are not yet available, i.e. before 2002, which agrees well with the AOD diversity for years 2000, 2008 and 2017 for all available satellite products (Fig. 3).

An attempt was performed to explain the offset in time series (AOD from the merged time series minus merged L3 AOD time series) obtained with different methods (Fig. 12) by looking at the AOD standard deviation between the available





products for particular years and at the number of products. However, no clear dependence was found, except that the negative offset above GCOS requirements (between -0.03 and -0.08) was observed before 2002 and positive offset (between 0.03 and 0.04) was observed mostly in 2002-2004 in some of the areas, as indicated in Fig. A9.

Thus, there is general consistency and similar temporal patterns between the two approaches (the merged time series and the timeseries from merged L3 AOD product), despite small differences, which are more pronounced at the beginning of the period, when less products are available.





**Figure 12. Time series of merged (approach 3, all aerosol types) annual time series AOD (blue) and annual AOD time series calculated from L3 merged (with approach 3, for all aerosol types) AOD product (olive) for the selected regions. Deviations between time series calculated with different statistics for all, background, fine-dominated and coarse-dominated aerosol are shown as ±1σ. For other regions, see Fig. A8.**



# 7 Regional and global annual, seasonal and monthly AOD merged time series

In Figs. 13-15 and Figs. A10-A12, we show the AOD time series obtained with different approaches – median for uncorrected AOD from 15 available products (approach 1), median for shifted to the ATSR ensemble AOD (approach 2), weighted time series (approach 3, all aerosol types) and time series from weighted AOD for annual, seasonal and monthly

products, respectively. We also show the $\pm1\sigma$ of AOD from all available products.

Median of the uncorrected AOD (approach 1) and merged (approach 3) AOD time series show the best agreement for all time scales (annual, seasonal, monthly). The deviation up to 0.05 ($AOD_{approach1} > AOD_{approach3}$) exists for Indonesia and North America before 2002, when both MODIS satellites become operational. For other regions, the deviation is considerably

lower (below 0.03). Median of shifted AOD (approach 2) has its clear limitations when the product chosen as a reference (ATSR ensemble, in our case) deviated considerably from other products in certain regions (e.g., AfN). In that case, offset exists between the time series merged with other approaches (1 and 3). However, this approach allows extending time series back to 1978-1994, where only the TOMS AOD long-term product currently exists and thus other merging approached are not applicable for that period.

Annual, seasonal and monthly time series from the merged (approach 3) L3 monthly AOD show a bit higher both signs deviation from time series discussed above. Interestingly, the seasonality is observed in the deviation. In AfN, the AOD from the monthly merged L3 is higher in autumn for the period of 1995-1999. In Bor and AsN (Figs. A10-A12), the deviation is higher in spring for the period of 1997-1999. The possible explanation might be due to the sparser coverage in those areas (due to restrictions in retrieval algorithms to retrieve bright surfaces, e.g., desert or snow).

Overall, good agreement exists among the time series calculated using different approaches. With only few percent exception, similar temporal patterns are reproduced and the offset between the AOD time series calculated with different approaches is within the GCOS requirement of 0.03 AOD.

There are of course caveats to these rather simple and straightforward merging approaches, which do not consider in much detail the differences in sampling and sensitivity to different conditions (e.g. surface brightness, number of independent

observables) of the different instruments and algorithms. It is well known that monthly, seasonal or annual gridded mean values carry large uncertainties, whether inferred from a few ground-based stations meant to represent a full grid cell, or from satellite images containing large gaps due to limited swath, clouds or failed retrievals. Pixel-level uncertainties are becoming available for a growing number of satellite products, and it would be highly beneficial if these estimated errors could be propagated consistently to those gridded monthly products. However, this requires deeper insight and new methods

to take into account correlation patterns among parts of the uncertainties, and to estimate practically the sampling-based uncertainties in light of approximated AOD variability. Altogether, as frequently requested from a user point of view, the stability and consistency of regional, merged AOD time series can be seen as strengthening our confidence in the reliability of satellite-based data records.



**Figure 13.** Annual AOD time series merged with three different approaches (blue, red, light blue for approaches 1-3, respectively) and AOD time series from the L3 merged data (approach 3, olive) for the selected regions. ±1σ of the AOD from all uncorrected AOD products is shown as light blue shadow. For other regions, see Fig. A10.

**Figure 14. Seasonal AOD time series merged with three different approaches (blue, red, light blue for approaches 1-3, respectively) and AOD time series from the L3 merged data (approach 3, olive) for the selected regions. ±1σ of the AOD from all uncorrected AOD products is shown as light blue shadow. For other regions, see Fig. A11.**



**Figure 15. Monthly AOD time series merged with three different approaches (blue, red, light blue for approaches 1-3, respectively) and AOD time series from the L3 merged data (approach 3, olive) for the selected regions. ±1σ of the AOD from all uncorrected AOD products is shown as light blue shadow. Note the different scale for ChinaSE, AfN and Ind. For other regions, see Fig. A12.**



**8 Conclusions**

This study has analysed the consistency of regional time records of monthly AOD from 15 different satellite products. These were obtained from a wide range of different instruments – TOMS, AVHRR, SeaWiFS, ATSR-2, AATSR, MODIS, MISR, POLDER, VIIRS and EPIC - with largely varying information content and sampling, and with different algorithms based on different remote sensing approaches, quality filtering, cloud masking and averaging.

Differences between those 15 data records in a set of regions with different characteristics across the globe were demonstrated and verified against a ground-based AERONET monthly mean gridded dataset in order to answer the question how well a satellite dataset can reproduce monthly gridded mean AERONET values in a region.

Regional AOD time series (monthly, seasonal, annual) from 15 different products (with different algorithms, measurement principles, number of independent observables, sampling) show good consistency of temporal patterns but significant biases due to all those differences.

To build the AOD data set merged from 15 different satellite products, three different merging approaches were introduced and tested. First, a simple median of the 15 uncorrected time records was conducted. Second, each record was shifted with a constant offset to one chosen record before the median was calculated. Third, the AERONET analysis was used to infer a ranking which was then used to calculate a weighted AOD mean. The third approach was applied to the AOD time series and to L3 monthly AOD product, later used to calculate regional time series. All merged regional AOD time series show the high consistency of temporal patterns and (where expected) between regions and the time records with their uncertainties (standard deviations shaded around the median values) are clearly able to illustrate the evolution of regional AOD. With few exceptions the three methods lead to very similar results (only the spread is much smaller and likely underestimated for the third approach), which is reassuring for the usefulness and stability of the merged products.

The AOD L3 monthly merged product (1995-2017) was developed and evaluated against the AERONET in the current paper. Evaluation results showed that the quality of the merged product is as good as that of the most highly ranked individual AOD products in each region. The corresponding time series are planned to be used in the AOD regional and global trend analysis, as well as in the inter-comparison with the modelled AOD product from the re-analysis.

**9 Data availability**

URL and doi (if available) of the products used in the current study are summarised in Table 2.

**Table 2**. URL and doi (if available) of the products used in the current study.

| Product | url/doi | archive | |
|---|---|---|---|





| TOMS | url | NASA's GES-DISC | https://disc.gsfc.nasa.gov/datasets?page=1&subject=Aerosols&measurement=Aerosol%20Optical%20Depth%2FThickness |
|---|---|---|---|
| OMI | url | NASA's GES-DISC | https://aura.gesdisc.eosdis.nasa.gov/data/Aura_OMI_Level2/OMAERUV.003/ |
| | doi | | 10.5067/Aura/OMI/DATA2004 |
| AVHRR | url | NASA NCCS | https://portal.nccs.nasa.gov/datashare/AVHRRDeepBlue |
| SeaWiFS | url | NASA GES DISC (via EarthData) | https://earthdata.nasa.gov/ |
| | doi | | 10.5067/MEASURES/SWDB/DATA304 |
| VIIRS | url | NASA LAADS (via EarthData) | https://earthdata.nasa.gov/ |
| | doi | | 10.5067/VIIRS/AERSDB_M3_VIIRS_SNPP.001 |
| ATSR ADV | url | ICARE | http://www.icare.univ-lille1.fr/archive |
| ATSR SU | url | ICARE | http://www.icare.univ-lille1.fr/archive |
| ATSR ensemble | url | ICARE | http://www.icare.univ-lille1.fr/archive |
| MODIS DT&DB * | url | NASA LAADS | https://ladsweb.modaps.eosdis.nasa.gov/ |
| | doi | | Terra: 10.5067/MODIS/MOD08_M3.061 Aqua: 10.5067/MODIS/MYD08_M3.061 |
| MODIS MAIAC | url | | https://search.earthdata.nasa.gov/search?q=MCD19&ok=MCD19 |
| MISR | url | | http://eosweb.larc.nasa.gov/project/misr/misr_table |
| | doi | | 10.5067/Terra/MISR/MIL3MAE_L3.004 |
| POLDER | url | ICARE | https://www.grasp-open.com http://www.icare.univ-lille1.fr |
| EPIC | url | | https://search.earthdata.nasa.gov/search?q=MCD19&ok=MCD19 |
| AOD merged | url | | will be available after the manuscript is accepted to ACP |
| AERONET | url | | https://aeronet.gsfc.nasa.gov/ |

\* Additional online documentation at: https://modis-atmos.gsfc.nasa.gov/, https://darktarget.gsfc.nasa.gov/,

https://deepblue.gsfc.nasa.gov/





## 10 Appendix



**Figure A1.** Upper line: summer AOD median for 2000 (*), 2008 and 2017 (X), calculated from the available products. Lines 2-6: AOD deviation of the different products from the annual median AOD for years 2000 (TOMS), 2017 (VIIRS and EPIC) or 2008 (all other products). AOD anomalies with respect to the AOD median are shown on the deviation plots. Global land and ocean AOD mean differences are shown for each product, when available. For annual offset, see Fig. 2.

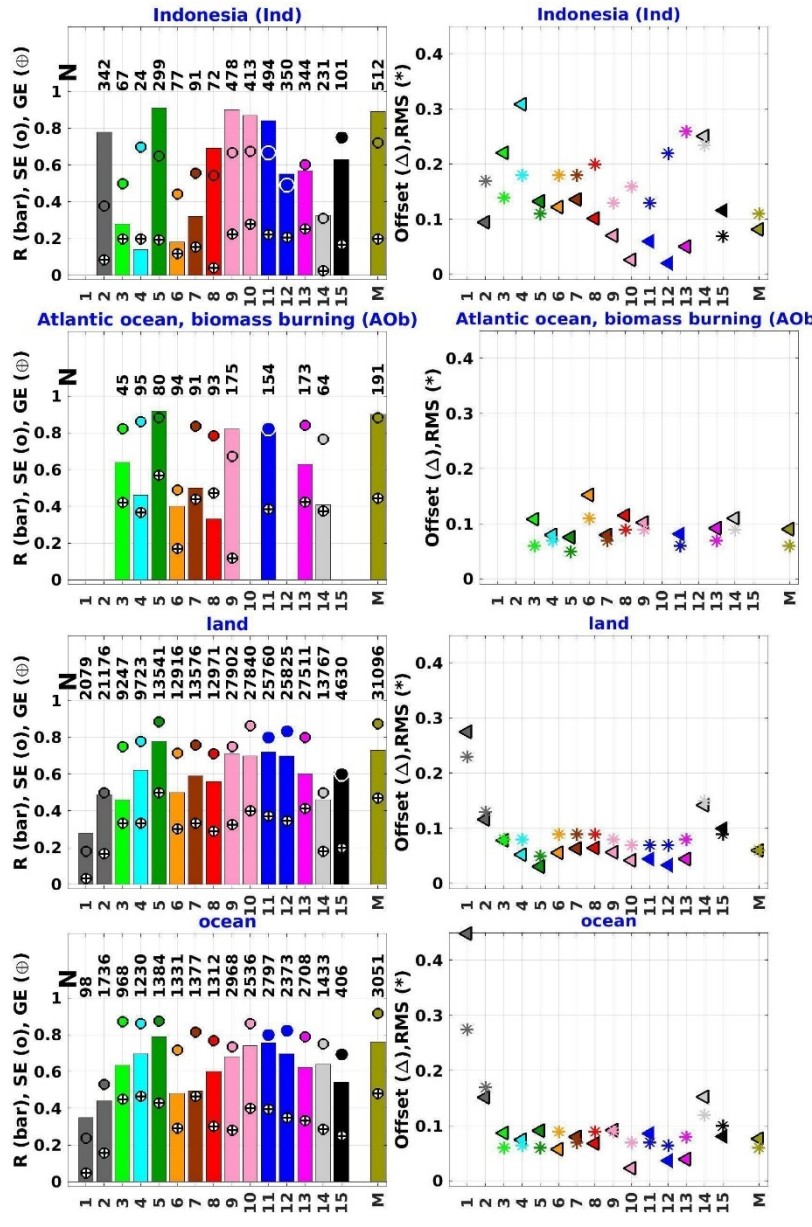





**Figure A2.** AERONET comparison statistics (correlation coefficient R, bar; fraction of pixels in the Spread Envelope, SE, ○; fraction of pixels satisfying the GCOS requirements, GE, ⊕; Offset, △; root mean square error RMS, *) for AOD monthly aggregates for each product (1:15, legend for products below the plot) and the L3 merged product (M, for details see Sect.6.2) with corresponding colors (legend) for the selected regions ( as in Fig. 1). N is a number of matches with AERONET.  Note, for products





**which do not provide global coverage (e.g., no retrieval over oceans), the results are missing. For Ind, AOb, land, ocean (upper panel), Bor, AsN, AsE, AsW, Aus (lower left panel), AfS, NAE, NAW, SA, Global (lower right panel). For other regions, see Fig. 5.**



**Figure A3. Annual AOD time series for the selected regions. For 2018, MODIS DT&DB and VIIRS AOD products are shown here but not used in the merging exercise. For other regions, see Fig. 6.**



**Figure A4. Monthly (J for January, F for February, etc., left plots) and seasonal (DJF, MAM, JJA, SON, middle plots) AOD time series for the selected regions for 2008, and yearly aggregate AOD for 2008 (Y, right) for selected regions. For other regions, see Fig. 7.**





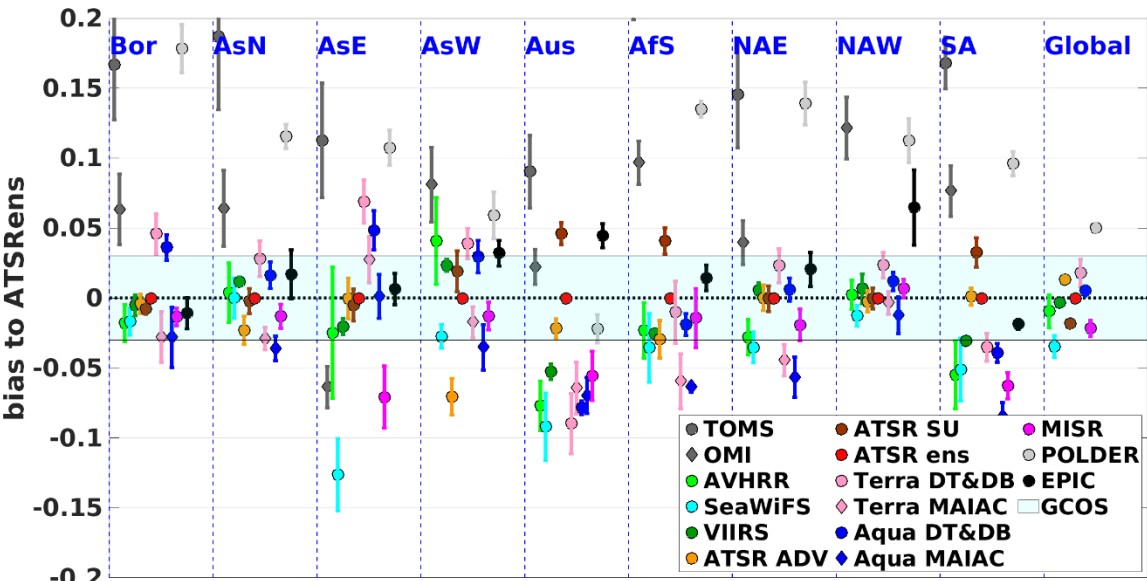

**Figure A5. Regional annual average AOD offset between each dataset and the ATSR_ensemble dataset. GCOS requirement for ±0.03 AOD is shown as a background color. For other regions, see Fig. 8.**





**Figure A6. Merged time series from uncorrected AODs (approach 1, blue) and from shifted AODs (approach 2, red) for the 8 selected regions (for both: mean, dotted line, median, solid line, and 1σ shadowed area). For other regions see Fig. 9.**





| 1 TOMS | 3 AVHRR | 6 ATSR_ADV | 9 Terra | 13 MISR |
|--------|---------|------------|---------|---------|
| 2 OMI  | 4 SeaWiFS | 7 ATSR_SU | 10 Terra MAIAC | 14 PARASOL POLDER |
|        | 5 VIIRS | 8 ATSR_ens | 11 Aqua | 15 EPIC MAIAC |
|        |         |            | 12 Aqua MAIAC | |

**Figure A7. Weight of each product for the selected regions for different aerosol types. For other regions, see Fig. 10.**





**Figure A8. Time series of merged (approach 3, all aerosol types) annual time series AOD (blue) and annual AOD time series calculated from L3 merged (with approach 3, for all aerosol types) AOD product (olive) for the selected regions. Deviations between time series calculated with different statistics for all, background, fine-dominated and coarse-dominated aerosol are shown as ±1δ. For other regions, see Fig. 12.**





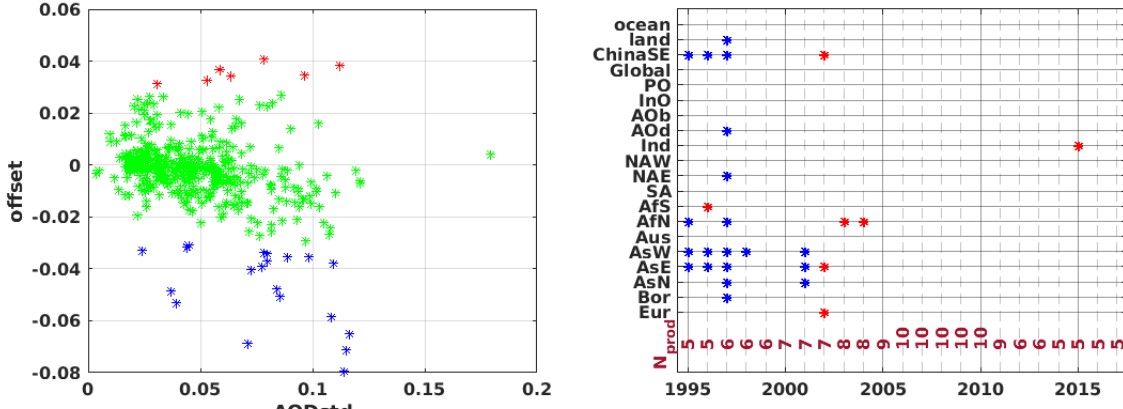

**Figure A9. Left: scatterplot for AOD standard deviations (AODstd) between the available products and offset between time series.**
**Green stars– offset within the GCOS requirements of 0.03, red and blue starts – positive and negative (respectively) offset outside**
**the GCOS requirements. Right – annual distribution of the offsets outside the GCOS requirements and number of the products**
**(Nprod) used for merging for each year.**





**Figure A10. Annual AOD time series merged with three different approaches (blue, red, light blue for approaches 1-3, respectively) and AOD time series from the L3 merged data (approach 3, olive) for the selected regions. ±1σ of the AOD from all uncorrected AOD products is shown as light blue shadow. For other regions, see Fig. 13.**

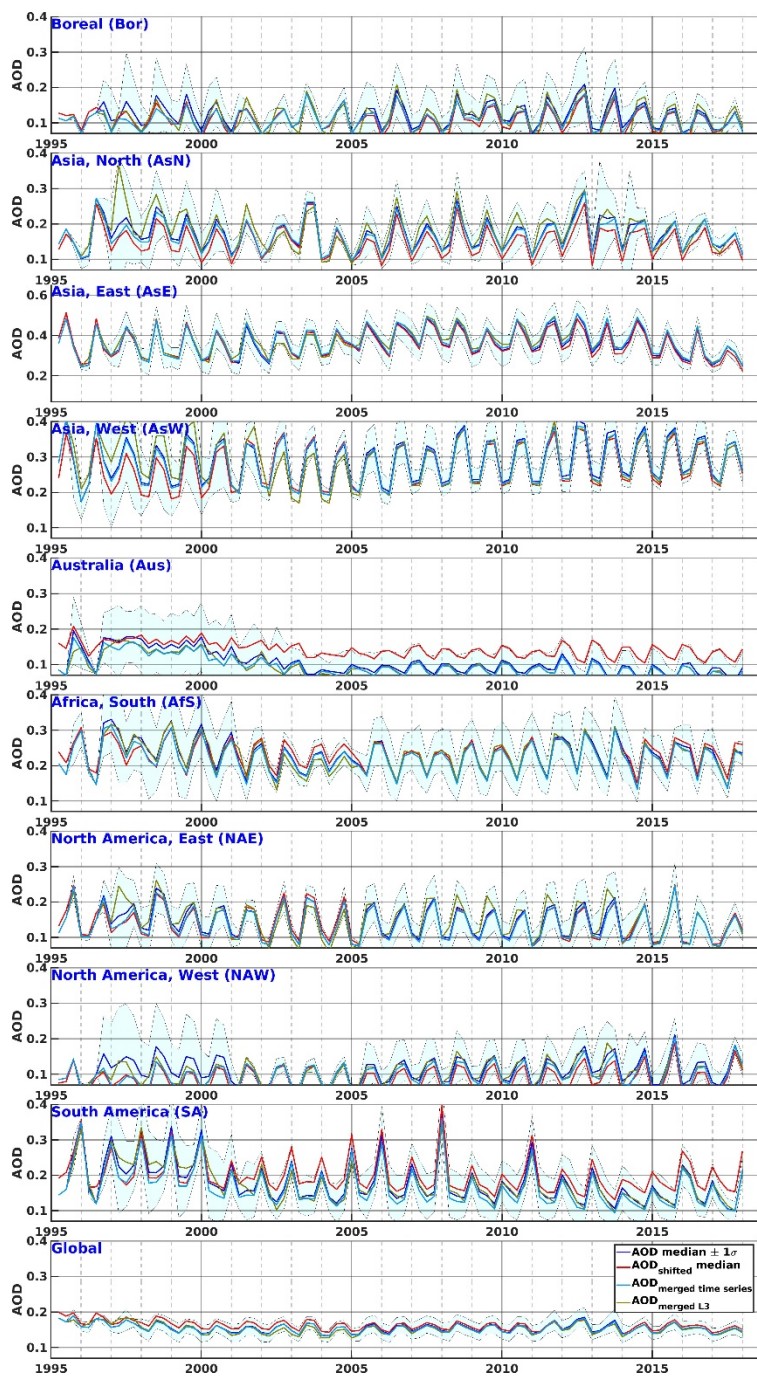

**Figure A11. Seasonal AOD time series merged with three different approaches (blue, red, light blue for approaches 1-3, respectively) and AOD time series from the L3 merged data (approach 3, olive) for the selected regions. ±1σ of the AOD from all uncorrected AOD products is shown as light blue shadow. For other regions, see Fig. 14.**

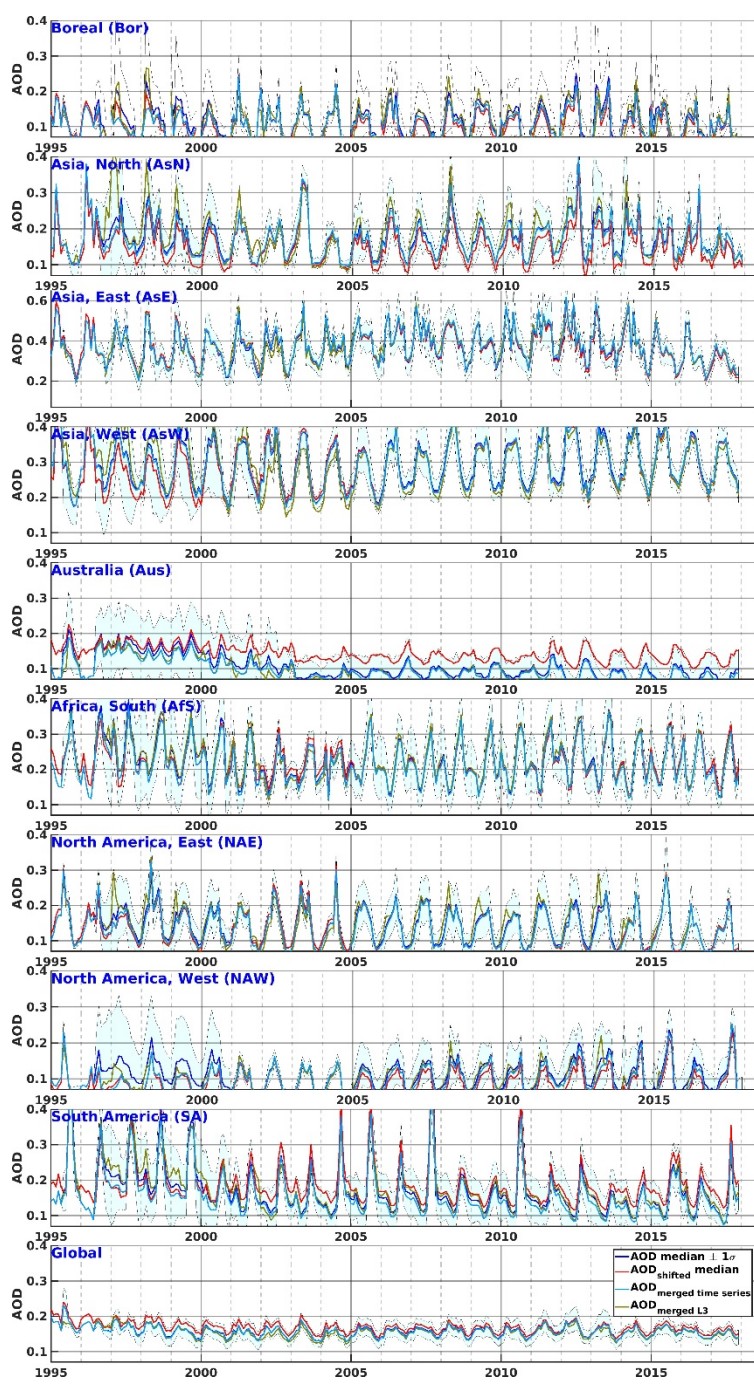

**Figure A12. Monthly AOD time series merged with three different approaches (blue, red, light blue for approaches 1-3, respectively) and AOD time series from the L3 merged data (approach 3, olive) for the selected regions. ±1σ of the AOD from all uncorrected AOD products is shown as light blue shadow. Note the different scale for ChinaSE, AfN and Ind. For other regions, see Fig. 15.**



## 11 Author contribution

The exercise on AOD merging has been initiated and widely discussed by the AeroCom/AeroSat community. The work has been performed by L. Sogacheva, who collected data, performed the analysis and wrote the extended draft of the manuscript. The evaluation results were widely discussed with the AOD data providers, who co-author the paper. Thomas Popp and Andrew M. Sayer considerably contributed to writing.

## 12 Acknowledgments

The authors thank attendees of AeroCom/AeroSat workshops over the past several years for lively and informative discussions, which helped provide the impetus for and shape this analysis. AeroCom and AeroSat are unfunded community networks which participants contribute to within the remit and constraints of their other aerosol research.

The work presented is partly supported by the Copernicus Climate Change Service (contracts C3S_312a_lot5 and C3S_312b_Lot2) which are funded by the European Union, with support from ESA as part of the Climate Change Initiative (CCI) project Aerosol_cci (ESA-ESRIN projects AO/1-6207/09/I-LG and ESRIN/400010987 4/14/1-NB) and the AirQast 776361 H2020-EO-2017 project.

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
