# Peer review of "Merging regional and global AOD records from major available satellite products"

_Atmospheric Chemistry and Physics, 2019_

## Referee Comment (RC1) · Anonymous Referee #2 · 23 Jul 2019

Review report of the ACP manuscript Merging regional and global AOD records from 15 available satellite products (https://doi.org/10.5194/acp-2019-446), by Sogacheva et al.

This manuscript discusses approaches to merge satellite AOD data sets from a large number of datasets derived from various instruments. The analyses start at the monthly AOD L3 products at a low spatial resolution (1x1° lat-lon). An extensive intercomparison of the various datasets is performed and different merging techniques are discussed.

The strongest part of this manuscript is the section 4, where the different datasets are compared. This could be a publication on its own. The weaker parts are the sections 5 and 6, which should be significantly improved in structure and readability.

[Figure]

Main comments One of the goals of the manuscript is to present a merged dataset. However, different merging methods are described, and no clear recommendation is made for a merged dataset. Also, a description of the final dataset is lacking. Therefore, the claim made in the abstract that a merged dataset is introduced is not fulfilled. If a dataset is presented its contents should be described, including on the technical level (in an appendix). Also, the dataset should be made available, preferably on one of the large datacentres, and with a doi.

The intended audience for the manuscript is not clear to me. If the intention is to describe a merged dataset, the intended reader is a potential user of that dataset. This user group is probably not an expert in the aerosol field and is probably not (so) interested in the performance of the individual underlying datasets (section 4, which is the largest part of the manuscript), but rather in a description of the performance and caveats of the merged one. This needs to be taken into account in the sections 5 and 6, which should be written at the right level, and more or less separate from section 4.

To summarize my main comments: - Make the merged datasets available and include a technical description. - Rewrite the sections 5 and 6 with the intended user of the dataset in mind as the audience. More comments on section 5 and 6 are found below. - Make clear what the final advertised merged data set is.

Section 5

This section should be rewritten, to clarify what was done and limited to methods that are used in the further analyses.

Section 5.1. This section is too brief and starts with a statement why the mean is not a good statistical indicator, whereas the it is one of the parameters that is calculated. What is missing is information on which data it is applied (to the monthly mean L3, or also to the seasonal and/or annual L3?).

Section 5.2. This section is too brief and unclear. With the information contained in

this section I would not be able to reproduce the results. The ATSR_ensemble is not available for the entire dataset. How do you deal with this? Clarify all the steps of the method.

Section 5.3. This section describes to methods: RM1 and RM2. However, RM1 is -as far as I can tell- not used in the rest of the paper. Therefore, it should be removed from this section, so approach 3 is limited to RM2 (in the remainder of the manuscript reference to RM2 should be changed to Approach 3). Furthermore, I propose to add one or more equations to clarify the procedure. Also, it should be clarified on which datasets it is applied, because if I understand section 6 correctly, there are also some sub-methods here (e.g. regional weights, monthly weights versus time-series weights, aerosol type weights).

Section 5.3 In approach 1 and 2 the mean, median and standard deviations are calculated. Why is this not done for approach 3 (see for example https://en.wikipedia.org/wiki/Weighted_median)?

Section 6 Section 6 should in my opinion describe the quality and caveats of the merged data using method 1, 2 and 3. It should not describe the performance of individual datasets. I think that part of my confusion seems to come from what is called the "merged product". As a reader, I think that methods 1,2 and 3 all yield a merged product but using different merging methods.

Section 6.1.1 I think this section doesn't belong in section 6. It describes the rationale for merging approach 2 and therefore should be moved to section 5.2.

Section 6.1.2: The title of this section is not covering the contents: in the current manuscript it is comparing the Merging Methods 1 and 2. However, I don't understand why Method 3 is left out in this section. Instead, I propose to describe the comparison of all three minutes, using figure 13 and to drop figures 9 and 12.

Section 6.2.1: This section described the weights; it doesn't assess the merged data

quality. I strongly suggest moving this section to section 5.3, which also increases the readability of that section.

Section 6.2.2: first paragraph. This describes sub-methods of approach 3 and should be described in section 5.3.

Figure 11, I like this figure, but why include it only for method 3? I think it should also be generated for methods 2 and 3 and the differences discussed.

Section 6.2.3. In the first 2 sentence 2 sub-methods are introduced of approach 3. This is not the right place, this should be done in section 5.3. The remainder of section 6.2.3 should be moved to 6.1.2, and also differences with the other methods should be described.

Section 7. I don't really see the need for this section. Line 1-22 would fit with the comparison of the time series of the three methods (e.g. 6.1.2). The last paragraph should be moved to the conclusion.

Specific comments

I strongly suggest adding a figure with timelines of the availability of all the products as part of the section 2. This information is also in Table 1, but a graphical overview would be a great help.

Table 1 presents the datasets, but the doi's and url's in Table 2. For each dataset the reference of doi (or url if doi is not available), should be included in Table 1, and the reference doi's and urls should be included in the list of references. Table 2 can be removed.

Page 14, lines 1-11. In the discussion of the comparison of AERONET with AOD L3 data, instead on the more common comparison with L2 data, one argument is missed. When L2 data is compared with AERONET with strict temporal and spatial criteria, the L2 data is implicitly cloud-cleared, because the AERONET data is only available under these conditions. This does not hold when comparing the L3 data. If the cloud clearing

is not optimal, this would lead to difference in the comparison results of L2-AERONET versus L3-AERONET.

Page 15, line 5 "manuscript" -> "work"

Page 16, line 16-17. It is not clear what is meant here. What does "different surface treatment" mean (compared to what?).

Page 19, section 4.3. Define how the ATSR_ensemble is computed.

Caption Figure 5, In light of my comments on section 5-6, I don't understand which merged product is shown as "M" in this figure.

Page 22, section 5.1. I would suggest to not only compute the standard deviation, but also percentiles, for example the 10th, 25th, 75th and 90th, because the standard deviation is very sensitive to outliers.

Page 22, line 5: "AOD weighted" is not clear. I suggest "Weighted mean, where the weights are derived from the comparisons with AERONET.

Page 23 line 26: "ATRS" should be "ATSR".

Page 29, line 1: "aerosol particles" should be "aerosol types"

---

## Referee Comment (RC2) · Anonymous Referee #3 · 29 Jul 2019

**Review report acp-2019-446**

The current study is divided in two parts. In the first one, an intercomparison among the most widely used aerosols datasets (obtained from spaceborne passive sensors) is performed while at the second one, various merging techniques are applied towards the development of a unified AOD product. To realize, monthly AODs derived by 15 satellite databases are analyzed over the period 1995-2017. The submitted manuscript is too long thus making difficult to the reader to get the information in a straightforward way. Moreover, the authors should make an effort to provide a better description of their merging methodology and their interpretation of the associated findings. Between the two parts, the first one (up to Section 4) can be a stand-alone paper without making substantial modifications while the second one (from Section 5) needs a lot of improvements regarding the description, interpretation and figures (e.g., add legends wherever do not exist, better description in the captions). Therefore, my recommendation to the authors is to split the current version of the manuscript in two separate works thus helping any potential reader to understand the overarching goal of the study as well as its components and the obtained scientific results. Below are listed my comments that should be addressed prior the publication of the submitted text.

1. **Page 4; Lines 14-16:** Is not clear what the authors want to say here.
2. **Page 8; Lines 2-3:** According to Table 1, there are not available data for 2000 from the AVHRR. Why don't you use 2001 in order to have full temporal coverage also from MODIS-Terra and MISR?
3. **Section 2:** Is there any criterion applied in the monthly products aiming at improving their quality (e.g., temporal representativeness, best quality retrievals) or just the raw products are utilized?
4. **Page 9; Lines 14-15:** Has any significance this threshold?
5. **Section 4.1:** It would be useful to add a table with the AOD averages over continental and maritime surfaces as well as for the whole globe.
6. **Page 10; Lines 26-27:** Where exactly? In the storm track zone (emission of marine aerosols due to strong winds) of the Southern Hemisphere or in the Southern Atlantic Ocean attributed to the transport of biomass aerosols from the central/south parts of Africa?
7. **Page 12; Lines 25-26:** Similar diversity levels are also encountered in the US, Mexico, S. America and Tibetan Plateau. Is there any explanation for that?
8. **Page 14; Lines 23-24:** The defined thresholds of Ångström exponent must be modified in order to create a buffer zone between fine and coarse aerosols modes. For example, fine and coarse particles can be "identified" when Ångström is higher and lower than 1.2 and 0.8, respectively. Even though the proposed limits are not the optimum, they are more realistic than the selected ones. An another solution could be the selection of representative AERONET stations for specific aerosol types or aerosol mixtures.
9. **Page 14; Lines 31-33:** I don't agree with the regional averaging of AERONET observations. Instead of giving equal weight on each AERONET site, it would be more correct (representative) to calculate the statistics on the whole AERONET dataset for each region.
10. **Page 15; Lines 3-4:** How has been defined the spread envelope? Why don't you use only the uncertainty limits defined by GCOS?
11. **Section 4.2.1:** The authors should guide better the reader by adding colors corresponding to aerosol groups in Figure 4. Also, rephrase the sentences in lines 13-15 and 25-27. In Figure 4, in the y-axis write that the difference is defined as satellite-AERONET, add a legend and rewrite the caption. Moreover, which is the background AOD? Are there available results for the total AOD without considering different aerosol classes?
12. **Figure 5:** Clarify that the offset is defined as satellite-AERONET.
13. **Page 22; Lines 8-10:** Rewrite this sentence because it is not clear.

14. **Section 5:** Definitely, a better and more analytical description of the applied merging approaches is needed explaining the benefits and the drawbacks of each methodology.
15. **Section 5.3:** In the RM2, why the levels are 10 and not 9 according to the discrimination of the computed statistics? For example, for the correlation coefficient they have been defined equal-range bins between 0.5 to 1 with a 0.05 step. If I have understood correctly this corresponds to 9 groups of R values instead of 10.
16. **Section 6:** The overarching goal of the current study (stated clearly in the title) is to merge different satellite databases. However, it is not clear to me which is the optimum methodology that should be followed. Also, I fully agree with the rearrangements proposed by the Reviewer #2.
17. **Page 23; Lines 26-27:** Please explain better this sentence.
18. **Figure 8:** Check if the shaded area corresponds to ±0.04.
19. **Section 7:** It is not clear why this Section is important.
20. **Page 37; Line 7:** What do you mean "*…AERONET monthly mean gridded dataset…*"?

---

## Author Comment (AC1) · 29 Nov 2019

We would like to thank the Reviewers for the thoughtful comments and suggestions. We have modified the manuscript according to these suggestions. Our replies are below (Reviewer's comments in *Italic*, response in normal face).

Other changes than those suggested by the Reviewers were applied to the manuscript during the revision.

The merging approaches have not been modified, but the number of the products used for merging was revised. TOMS, OMI and EPIC products, which were reported at other than 0.55 µm, were removed from merging. However, we keep those products in the inter-comparison and evaluation with AERONET exercises.

Another product, AVHRR NOAA (over ocean) was added.

In the merging approach 1, the reference to estimate the average offsets with individual products was re-considered: ATSR_ens was replaced with Terra DT&DB. With this change, the overlapping period exists between the reference and all individual products, thus the direct inter-comparison is possible (in the version submitted to the ACPD the offsets between ATSR_ens and VIIRS and ERIC were calculated in two steps, with estimating intermediate offsets to MODIS Aqua).

Section on the estimation of uncertainties in the L3 merged AOD product was added; the spatial and temporal uncertainties are shown and discussed.

In Sect. 6 (revised version), when we discuss the results from different methods for merging annual time series, we now show TOMS (over land) and AVHRR NOAA (over ocean), both shifted to the merged time series (shifted to the reference in the version submitted to the ACPD).

We thoroughly revised the paper, which required an input from a new co-author.
*Review report of the ACP manuscript Merging regional and global AOD records from 15 available satellite products (https://doi.org/10.5194/acp-2019-446), by Sogacheva et al.*

*This manuscript discusses approaches to merge satellite AOD data sets from a large number of datasets derived from various instruments. The analyses start at the monthly AOD L3 products at a low spatial resolution (1x1◦ lat-lon). An extensive intercomparison of the various datasets is performed and different merging techniques are discussed.*

*The strongest part of this manuscript is the section 4, where the different datasets are compared. This could be a publication on its own. The weaker parts are the sections 5 and 6, which should be significantly improved in structure and readability.*

*Main comments*

*One of the goals of the manuscript is to present a merged dataset. However, different merging methods are described, and no clear recommendation is made for a merged*

*dataset. Also, a description of the final dataset is lacking. Therefore, the claim made in the abstract that a merged dataset is introduced is not fulfilled. If a dataset is presented its contents should be described, including on the technical level (in an appendix). Also, the dataset should be made available, preferably on one of the large datacentres, and with a doi.*

According to the Reviewer's comments, we revised considerably sections 5-7, where methods for merging are described. The scheme for the merging approaches was added to the introduction for merging approaches (Sec. 4 in the revised version).

All merged products are now described and validated (Section 5 in the revised version). Section on the pixel-level uncertainties for the final L3 merged products is added. The recommendations are given on the final merged product. The merged data set will be openly available at Finnish National Satellite Data center, http://nsdc.fmi.fi/ ; the full link will be provided in the manuscript accepted for publication.

*The intended audience for the manuscript is not clear to me. If the intention is to describe a merged dataset, the intended reader is a potential user of that dataset. This user group is probably not an expert in the aerosol field and is probably not (so) interested in the performance of the individual underlying datasets (section 4, which is the largest part of the manuscript), but rather in a description of the performance and caveats of the merged one. This needs to be taken into account in the sections 5 and 6, which should be written at the right level, and more or less separate from section 4.*

So far, individual products, which have certain limitations discussed in the manuscript, have been used in the air quality and climate studies. We expect that the potential users, if not experts, have a good knowledge on aerosols. For the merged product offered here, which has the main advantage in better temporal overage with similar or better quality, we expect the same audience, which use other individual satellite products, but since the merged product allows looking at the longer period, the climate researchers will benefit from having access to the longer data set.
The interest to the AOD merged product was shown by the AeroCom community. Several request from modelers have already been obtained for evaluation of the modelled AOD products.
To make the manuscript useful for experts in different fields, the discussion of the performance of the merged products has been considerably enlarged by including the evaluation results for all tested merged product and inter-comparison of the selected merged product with individual products.

***To summarize my main comments:***
 *- Make the merged datasets available and include a technical description.*
 *- Rewrite the sections 5 and 6 with the intended user of the dataset in mind as the audience. More comments on section 5 and 6 are found below.*
 *- Make clear what the final advertised merged data set is.*

The technical description has been revised and supported by the results (Section 5 and 6 are combined).

The scheme for the merging approaches was added to the introduction for merging approaches (Sec. 4 in the revised version).

New section, there the merged L3 monthly products are introduced, evaluated, inter-compared is added.

The main merged product is chosen and inter-compared with individual products.

Section on the pixel-level uncertainties for the final L3 merged products is added.

**Section 5**

*This section should be rewritten, to clarify what was done and limited to methods that are used in the further analyses.*

Section 5 and 6 are combined. The technical description has been revised and supported by the results

*Section 5.1. This section is too brief and starts with a statement why the mean is not a good statistical indicator, whereas the it is one of the parameters that is calculated. What is missing is information on which data it is applied (to the monthly mean L3, or also to the seasonal and/or annual L3?).*

The "mean" approach has been removed from the manuscript.

It was clarified in the text, that the merging has been applied to L3 monthly dataset and annual/seasonal/monthly time series.

*Section 5.2. This section is too brief and unclear. With the information contained in this section I would not be able to reproduce the results. The ATSR_ensemble is not available for the entire dataset. How do you deal with this? Clarify all the steps of the method.*

Section 5.2 was combined with Section 5.1. The offset correction method was supported by the offset correction results.

*Section 5.3. This section describes to methods: RM1 and RM2. However, RM1 is -as far as I can tell- not used in the rest of the paper. Therefore, it should be removed from this section, so approach 3 is limited to RM2 (in the remainder of the manuscript reference to RM2 should be changed to Approach 3). Furthermore, I propose to add one or more equations to clarify the procedure. Also, it should be clarified on which datasets it is applied, because if I understand section 6 correctly, there are also some sub-methods here (e.g. regional weights, monthly weights versus time-series weights, aerosol type weights).*

In the revised version, results for RM2 are shown and inter-compared with the results from other merging approaches.

The equation for the weighted mean is added.

The datasets, on which the approaches and sub-methods are applied, is discussed in the new Sect. 5, where the merged L3 products are introduced, evaluated, and inter-compared.

*Section 5.3 In approach 1 and 2 the mean, median and standard deviations are calculated. Why is this not done for approach 3 (see for example https://en.wikipedia.org/wiki/Weighted_median)?*

The conclusion on the choice of one final merged product is based on the fact, that there is a very small deviation between L3 products and time series merged with different approached (1 or 2) and sub-methods (aerosol types), except for approach 1 applied to the shifted products. Thus, we do not see the need to calculated average (median) from all tested merged product, which, if done, makes the analysis more complicated with no significant improvement.

**Section 6**
*Section 6 should in my opinion describe the quality and caveats of the merged data using method 1, 2 and 3. It should not describe the performance of individual datasets. I think that part of my confusion seems to come from what is called the "merged product". As a reader, I think that methods 1,2 and 3 all yield a merged product but using different merging methods.*

In Sect.6 (ACPD version), which is combined with Sect.5 in the revised version, it is important, in our opinion, to show the results for individual products because those results contribute to the final ranking of the products. In the revised version, the results are shown for two regions only, Europe and ChinaSE, as an example. The results for all regions are moved to Supplement.

*Section 6.1.1 I think this section doesn't belong in section 6. It describes the rationale for merging approach 2 and therefore should be moved to section 5.2.*

Section 6.1.1 was moved to Section 4.1 in the revised version

*Section 6.1.2: The title of this section is not covering the contents: in the current manuscript it is comparing the Merging Methods 1 and 2. However, I don't understand why Method 3 is left out in this section. Instead, I propose to describe the comparison of all three minutes, using figure 13 and to drop figures 9 and 12.*

Approaches 1 and 2, as in the ms submitted to the ACPD, are combined in the revised version. Annual time series, calculated with the approach 1 (uncorrected AOD) and approach 2 (weighted AOD, former approach 3) are now introduced in Sect. 6; differences in the results from different approaches and different steps (time series from merged L3 product and merged time series) are discussed.

*Section 6.2.1: This section described the weights; it doesn't assess the merged data quality. I strongly suggest moving this section to section 5.3, which also increases the readability of that section.*

Section 6.2.1 was moved to Sect.4.2, where both method and results (weights) are discussed

*Section 6.2.2: first paragraph. This describes sub-methods of approach 3 and should be described in section 5.3.*
*Figure 11, I like this figure, but why include it only for method 3? I think it should also be generated for methods 2 and 3 and the differences discussed.*

In the revised version, the evaluation results for all products are shown and discussed (Sect. 5.1). The evaluation results for the chosen product (former Fig.11) are summarized in Sec. 5.2.1. Uncertainties for the final merged product (M) are introduced and discussed in Sect.5.2.2

*Section 6.2.3. In the first 2 sentence 2 sub-methods are introduced of approach 3. This is not the right place, this should be done in section 5.3. The remainder of section 6.2.3 should be moved to 6.1.2, and also differences with the other methods should be described.*

Section 6.2.3 was partly combined with Sect.4 (revised version) and partly moved to Sect.6 (revised version). The differences in the time series merging results are discussed in Sect 6; results are summarized in Table 3.

*Section 7. I don't really see the need for this section. Line 1-22 would fit with the comparison of the time series of the three methods (e.g. 6.1.2). The last paragraph should be moved to the conclusion.*

Section 7 (Sect.6 in the revised version) has been considerably revised. The merged time series are introduced, the difference between them is discussed.
The last paragraph was moved to the conclusion.

***Specific comments***
*I strongly suggest adding a figure with timelines of the availability of all the products as part of the section 2. This information is also in Table 1, but a graphical overview would be a great help.*

The information on the availability of the products use for merging is now summarized in Table 2. The availability of the other products (TOMS, OMI, AVHRR NOAA) is mentioned in the text.

*Table 1 presents the datasets, but the doi's and url's in Table 2. For each dataset the reference of doi (or url if doi is not available), should be included in Table 1, and the*

*reference doi's and urls should be included in the list of references. Table 2 can be removed.*

According to the ACP rule, that data availability should be included as a separate section. In the title for Table 1, we added the information that the data availability is summarised in Table 4.

*Page 14, lines 1-11. In the discussion of the comparison of AERONET with AOD L3 data, instead on the more common comparison with L2 data, one argument is missed.*

We discuss the L3 validation in Lines 3-11. To make it more clear, we now mention L3 specifically.

*When L2 data is compared with AERONET with strict temporal and spatial criteria, the L2 data is implicitly cloud-cleared, because the AERONET data is only available under these conditions. This does not hold when comparing the L3 data. If the cloud clearing is not optimal, this would lead to difference in the comparison results of L2-AERONET versus L3-AERONET.*

The problems related to the difference in cloud screening are mentioned in abstract, Introduction, Sect.2 and conclusions, briefly or with some details.

*Page 15, line 5 "manuscript" -> "work"*

Corrected

*Page 16, line 16-17. It is not clear what is meant here. What does "different surface treatment" mean (compared to what?).*

Different approaches for surface treatment in different products. Clarification is added.

*Page 19, section 4.3. Define how the ATSR_ensemble is computed.*

The definition for the ATSR_ensmble product is added to Sect. 2.2

*Caption Figure 5, In light of my comments on section 5-6, I don't understand which merged product is shown as "M" in this figure.*

The caption was revised by including the reference to the merged product

*Page 22, section 5.1. I would suggest to not only compute the standard deviation, but also percentiles, for example the 10th, 25th, 75th and 90th, because the standard deviation is very sensitive to outliers.*

The standard deviations were low, thus the contribution of the outliers, if existed, was negligible.

*Page 22, line 5: "AOD weighted" is not clear. I suggest "Weighted mean, where the weights are derived from the comparisons with AERONET.*

The whole paragraph was re-written

*Page 23 line 26: "ATRS" should be "ATSR".*

Corrected, as suggested by the Reviewer

*Page 29, line 1: "aerosol particles" should be "aerosol types"*

Corrected, as suggested by the Reviewer

---

## Author Comment (AC2) · 29 Nov 2019

We would like to thank the Reviewers for the thoughtful comments and suggestions. We have modified the manuscript according to these suggestions. Our replies are below (Reviewer's comments in *Italic*, response in normal face).

Other changes than those suggested by the Reviewers were applied to the manuscript during the revision.

The merging approaches have not been modified, but the number of the products used for merging was revised. TOMS, OMI and EPIC products, which were reported at other than 0.55 µm, were removed from merging. However, we keep those products in the inter-comparison and evaluation with AERONET exercises.

Another product, AVHRR NOAA (over ocean) was added.

In the merging approach 1, the reference to estimate the average offsets with individual products was re-considered: ATSR_ens was replaced with Terra DT&DB. With this change, the overlapping period exists between the reference and all individual products, thus the direct inter-comparison is possible (in the version submitted to the ACPD the offsets between ATSR_ens and VIIRS and ERIC were calculated in two steps, with estimating intermediate offsets to MODIS Aqua).

Section on the estimation of uncertainties in the L3 merged AOD product was added; the spatial and temporal uncertainties are shown and discussed.

In Sect. 6 (revised version), when we discuss the results from different methods for merging annual time series, we now show TOMS (over land) and AVHRR NOAA (over ocean), both shifted to the merged time series (shifted to the reference in the version submitted to the ACPD).

We thoroughly revised the paper, which required an input from a new co-author.

**Review report #3, acp-2019-446**

*The current study is divided in two parts. In the first one, an intercomparison among the most widely used aerosols datasets (obtained from spaceborne passive sensors) is performed while at the second one, various merging techniques are applied towards the development of a unified AOD product. To realize, monthly AODs derived by 15 satellite databases are analyzed over the period 1995-2017. The submitted manuscript is too long thus making difficult to the reader to get the information in a straightforward way. Moreover, the authors should make an effort to provide a better description of their merging methodology and their interpretation of the associated findings. Between the two parts, the first one (up to Section 4) can be a stand-alone paper without making substantial modifications while the second one (from Section 5) needs a lot of improvements regarding the description, interpretation and figures (e.g., add legends wherever do not exist, better description in the captions). Therefore, my recommendation to the authors is to split the current version of the manuscript in two separate works thus helping any potential reader to understand the overarching goal of the study as well as its components and the obtained scientific results. Below are listed my comments that should be addressed prior the publication of the submitted text.*

The main reason for keeping those two parts of the analysis together is that the evaluation results for the individual products are used for merging. To shorten "part 1", Sect. 4.1 (AOD spatial distribution and diversity) and 4.4 (AOD annual cycles) and some figures from other sections were moved to Supplement; the discussion on the quality of the individual products was also shortened a bit.

**1.** *Page 4; Lines 14-16: Is not clear what the authors want to say here.*

The sentence was replaced with "Whereas a lack of diversity among data sets does not mean that they have converged on the true value e.g., AErosol RObotic NETwork (AERONET, Holben et al., 1998) AOD, which is a recognized standard for instantaneous AOD reference, the existence of unexplained diversity does imply they have not."

2. *Page 8; Lines 2-3: According to Table 1, there are not available data for 2000 from the AVHRR. Why don't you use 2001 in order to have full temporal coverage also from MODIS-Terra and MISR?*

TOMS is not reliable in Nov-Dec 2001. Thus, year 2000 was chosen.

3. *Section 2: Is there any criterion applied in the monthly products aiming at improving their quality (e.g., temporal representativeness, best quality retrievals) or just the raw products are utilized?*

Monthly data from the open sources or obtained from the data providers was utilized, except for MISR and AVHRR NOAA (included into analysis during the revision), for which the monthly products were reported at lower resolution. For those, a simple averaging to 1° was applied to match the other products (details in Sect. 2.2). Most of the products do not include the quality flags.

4. *Page 9; Lines 14-15: Has any significance this threshold?*
In the revised version, the actual number of the maxima offset (0.011) is given.

5. *Section 4.1: It would be useful to add a table with the AOD averages over continental and maritime surfaces as well as for the whole globe.*

The absolute median AOD numbers for land/ocean/globe for years 2000, 2008 and 2017 are in the upper panel of Fig.2; Offsets from global/land/ocean averaged AOD is given for all individual products, when available. Thus, the actual AOD for each product can be easily calculated.

6. *Page 10; Lines 26-27: Where exactly? In the storm track zone (emission of marine aerosols due to strong winds) of the Southern Hemisphere or in the Southern Atlantic Ocean attributed to the transport of biomass aerosols from the central/south parts of Africa?*

We added a short discussion on the possible contribution of the storm track zone to the elevated AOD over Southern Ocean and provided the reference.

7. *Page 12; Lines 25-26: Similar diversity levels are also encountered in the US, Mexico, S. America and Tibetan Plateau. Is there any explanation for that?*

In S. America (Amazon), the difficulties/differences in cloud screening might be an issue. Big events of the forest fires might be screened as cloud in some products. To check that, L2 AOD and cloud screening results should be intercompared, which is out of the scope of that manuscript. Same for thick dust events. Different assumptions in bright surface treatment might cause another

offset in AOD. However, the AOD diversity is changing there, related to the time period and availability of the products, while over Australia the deviation remains constant along the time.

8. *Page 14; Lines 23-24: The defined thresholds of Ångström exponent must be modified in order to create a buffer zone between fine and coarse aerosols modes. For example, fine and coarse particles can be "identified" when Ångström is higher and lower than 1.2 and 0.8, respectively. Even though the proposed limits are not the optimum, they are more realistic than the selected ones. An another solution could be the selection of representative AERONET stations for specific aerosol types or aerosol mixtures.*

First, we wanted to be consistent with previous studies (Sayer al., 2018a, Sogacheva et al., 2018 a, b). The other reason is that in monthly aggregates the aerosol types are defined as a median from the 1-month period, while the presence of other types is possible and often obvious. Thus, such a strict differentiation of the aerosol types, suggested by the Reviewer, is not of great importance in the current study.
We included to the Supplement the figure (Fig. S5), where for each AERONET station the prevailing annual and seasonal aerosol type has been estimated based on the chosen criteria. There is a sense in the results obtained, which confirms the applicability of the aerosol type classification suggested in the current study. The results also show that aerosol types differ from one Aeronet stations to another in the same region, thus the prevailing aerosol type can't be defined with a high confidence for the chosen regions.

9. *Page 14; Lines 31-33: I don't agree with the regional averaging of AERONET observations. Instead of giving equal weight on each AERONET site, it would be more correct (representative) to calculate the statistics on the whole AERONET dataset for each region.*

We tested the approach suggested by the Reviewer, when the study was planned. The validation results for specific areas were often similar with two approaches. However, following the logic that the weight of the validation results might be biased toward the longest time series from a few AERONET stations in the particular area, which are not fully representative for the big region, we chose the other validation approach, explained in the manuscript.

10. *Page 15; Lines 3-4: How has been defined the spread envelope? Why don't you use only the uncertainty limits defined by GCOS?*

The results for the spread envelope, defined in Sect 4.2 of the version submitted to ACPD, are removed in the revised version.

11. *Section 4.2.1: The authors should guide better the reader by adding colors corresponding to aerosol groups in Figure 4. Also, rephrase the sentences in lines 13-15 and 25-27. In Figure 4, in the y-axis write that the difference is defined as satellite-AERONET, add a legend and rewrite the caption. Moreover, which is the background AOD? Are there available results for the total AOD without considering different aerosol classes?*

The legend with the explanation for the colours was added

The explanation to the background AOD was given in the text and now added to the figure caption. The evaluation and the following merging were performed also for all aerosol types (total AOD).

12. **Figure 5:** *Clarify that the offset is defined as satellite-AERONET.*

Clarification was added to the text and y-label caption.

13. **Page 22; Lines 8-10:** *Rewrite this sentence because it is not clear.*

The whole paragraph was revised.

14. **Section 5:** *Definitely, a better and more analytical description of the applied merging approaches is needed explaining the benefits and the drawbacks of each methodology.*

The scheme for the merging approaches was added in the introduction for merging approaches (Sec. 4 in the revised version).
The description of the applied merging approaches has been expanded and supported by further discussion of the results.
Section on the pixel-level uncertainties for the final L3 merged product is added.

15. **Section 5.3:** *In the RM2, why the levels are 10 and not 9 according to the discrimination of the computed statistics? For example, for the correlation coefficient they have been defined equalrange bins between 0.5 to 1 with a 0.05 step. If I have understood correctly this corresponds to 9 groups of R values instead of 10.*
With the 0.05 step, 10 bins (groups) exist between 0.5 and 1

| 1 | 0.50 | 0.55 |
|----|------|------|
| 2 | 0.55 | 0.60 |
| 3 | 0.60 | 0.65 |
| 4 | 0.65 | 0.70 |
| 5 | 0.70 | 0.75 |
| 6 | 0.75 | 0.80 |
| 7 | 0.80 | 0.85 |
| 8 | 0.85 | 0.90 |
| 9 | 0.90 | 0.95 |
| 10 | 0.95 | 1.00 |

16. **Section 6:** *The overarching goal of the current study (stated clearly in the title) is to merge different satellite databases. However, it is not clear to me which is the optimum methodology that should be followed. Also, I fully agree with the rearrangements proposed by the Reviewer #2.*

The manuscript has been revised considerably by adding the results from the intercompariosn between the products merged with different approached and considering different aerosol types. Based in the inter-comparison results, one merged product was chosen. The summarised

validation results for that product are shown in the new section, which also now includes the inter-comparison between the merged and individual products.

17. **Page 23; Lines 26-27:** *Please explain better this sentence.*

Terra DT&DB was chosen as a reference for offset correction in the revised manuscript. The text was revised accordingly.

18. *Figure 8: Check if the shaded area corresponds to ±0.04.*

Checked. The shaded area corresponds to ±0.03

19. *Section 7: It is not clear why this Section is important.*

In the revised version, the merged annual/seasonal/monthly time series are introduced in Sect.6. Difference between time series merged with different approaches is discussed.

20. *Page 37; Line 7: What do you mean "…AERONET monthly mean gridded dataset…"?*

"gridded" is removed.

---

## Author Response (AR2)

Dear Stelios Kazadzis,

Thank you for reviewing and accepting our manuscript for publication in ACP.

The last language editing was done by Andy Sayer.

Best regards,

Larisa Sogacheva